

# Global source attribution of sulfate concentration, direct and
# indirect radiative forcing
Yang Yang[1*], Hailong Wang[1*], Steven J. Smith[2], Richard Easter[1], Po-Lun Ma[1], Yun
Qian[1], Hongbin Yu[3], Can Li[3,4], Philip J. Rasch[1]
[1]Atmospheric Science and Global Change Division, Pacific Northwest National
Laboratory, Richland, Washington, USA
[2]Joint Global Change Research Institute, Pacific Northwest National Laboratory,
College Park, Maryland, USA
[3]NASA Goddard Space Flight Center, Greenbelt, Maryland, USA
[4]Earth System Science Interdisciplinary Center, University of Maryland, College Park,
Maryland, USA
*Correspondence to yang.yang@pnnl.gov and hailong.wang@pnnl.gov



**Abstract**

The global source-receptor relationships of sulfate concentration, direct and
indirect radiative forcing (DRF and IRF) from sixteen regions/sectors for years
2010-2014 are examined in this study through utilizing a sulfur source-tagging
capability implemented in the Community Earth System Model (CESM) with winds
nudged to reanalysis data. Sulfate concentrations are mostly contributed by local
emissions in regions with high emissions, while over regions with relatively low $SO_2$
emissions, the near-surface sulfate concentrations are primarily attributed to non-local
sources from long-range transport. The export of $SO_2$ and sulfate from Europe
contributes 16–20% of near-surface sulfate concentrations over North Africa,
Russia/Belarus/Ukraine (RBU) region and Central Asia. Sources from the Middle
East account for 15–24% of sulfate over North Africa, Southern Africa and Central
Asia in winter and autumn, and 19% over South Asia in spring. Sources in RBU
account for 21–42% of sulfate concentrations over Central Asia. East Asia accounts
for about 50% of sulfate over Southeast Asia in winter and autumn, 15% over RBU in
summer, and 11% over North America in spring. South Asia contributes to 11–24% of
sulfate over Southeast Asia in winter and spring. Regional source efficiencies of
sulfate concentrations are higher over regions with dry atmospheric conditions and
less export, suggesting that lifetime of aerosols, together with regional export, is
important in determining regional air quality. The simulated global total sulfate DRF is
$-0.42$ W m$^{-2}$, with $-0.31$ W m$^{-2}$ contributed by anthropogenic sulfate and $-0.11$ W m$^{-2}$
contributed by natural sulfate, relative to a state with no sulfur emissions. In the



Southern Hemisphere tropics, dimethyl sulfide (DMS) contributes 17–84% to the total
DRF. East Asia has the largest contribution of 20–30% over the Northern Hemisphere
mid- and high-latitudes. A 20% perturbation of sulfate and its precursor emissions
gives a sulfate incremental IRF of −0.44 W m$^{-2}$. DMS has the largest contribution,
explaining −0.23 W m$^{-2}$ of the global sulfate incremental IRF. Incremental IRF over
regions in the Southern Hemisphere with low background aerosols is more sensitive to
emission perturbation than those over the polluted Northern Hemisphere.



## 1. Introduction

Sulfate is an important aerosol that poses health risks (Fajersztajn et al., 2013;
Xu et al., 2013; Peplow, 2014) and sulfur deposition is a major driver of ecosystem
acidification (Driscoll et al., 2010). Due to long-range transport, local sulfate pollution
could result from intercontinental influences, making domestic efforts of improving air
quality inefficient (Part et al., 2004; Bergin et al., 2005; Liu and Mauzerall, 2007). In
addition, sulfate aerosol substantially perturbs the radiation budget of the Earth
directly through scattering incoming solar radiation and indirectly through modifying
cloud microphysical properties (Lohmann and Feichter, 2005; Stevens and Feingold,
2009; Myhre et al., 2013). On a global average basis, anthropogenic sulfate aerosol
contributes a negative direct radiative forcing (DRF) of $-0.4 \pm 0.2$ W m$^{-2}$ (Boucher et
al., 2013). The negative radiative forcing from sulfate partly offsets the positive
radiative forcing from greenhouse gases. Therefore, accurate understanding of
source attribution of sulfate and its radiative forcing is important for both regional air
quality and global climate mitigation (Shindell et al., 2012), which are of great interest
to not only science community but also the general public and policymakers.
Sulfate aerosol is produced through oxidation of sulfur dioxide ($SO_2$) by the
hydroxyl radical (OH) in gas phase and aqueous phase oxidation mainly by hydrogen
peroxide ($H_2O_2$) (Martin and Damschen, 1981). The $SO_2$ precursor is mainly emitted
from fossil-fuel combustion (Lu et al., 2010). In recent decades, $SO_2$ emissions from
many developing countries in East Asia and South Asia have increased substantially
as a result of accelerated urbanization and rapid economic growth (Streets et al.,



2000; Pham et al., 2005). In contrast, due to air pollution regulations, $SO_2$ emissions
in North America and Europe have decreased significantly since 1980–1990 (Smith
et al., 2011; Prechtel et al., 2001). As a consequence, source attribution of sulfate has
changed with time over recent decades.

Previous studies have reported that regional aerosols, including sulfate, are

produced not only by domestic emissions, but also by distant sources through
long-range transport (Jacob et al., 2003; Jaffe et al., 2003; Park et al., 2004; Heald et
al., 2006; Liu et al., 2008; Liu et al., 2009; Yu et al., 2012). For example, the strong
anthropogenic emissions over East Asia have led to an increasing interest in
quantifying the impact of aerosols exported from East Asia. Recent studies indicate
that the transpacific transport of sulfate from East Asia contributes to 30–50% of the
background (sulfate produced from non-local emissions) surface concentrations in
the Western U.S. and 10–30% in the Eastern U.S. (Park et al., 2004; Hadley et al.,
2007; Liu et al., 2008), which are larger than contributions from all other foreign
sources (Liu et al., 2009). In addition, among the major emitting regions assessed for
2001 conditions, European sources were shown to account for 1–5 µg m$^{-3}$ of surface
sulfate concentration over northern Africa and western Asia, and their contribution to
East Asia (0.2–0.5 µg m$^{-3}$) was twice as much as the contribution (0.1–0.2 µg m$^{-3}$) of
Asian sources to North America (Chin et al., 2007).

Due to the important role of sulfate aerosol in the climate system, knowing the

relative significance of sulfate radiative forcing from different source regions is useful
for climate mitigation. Some previous studies examined the impact of emission





reductions on global and regional DRF and the influence of long-range transport (Yu
et al., 2013; Bellouin et al., 2016; Stjern et al., 2016). Yu et al. (2013) examined
changes in aerosol DRF resulting from a 20% reduction in anthropogenic emissions
from four major polluted regions (namely North America, Europe, East Asia, and
South Asia) in Northern Hemisphere, using simulations by nine models from the first
phase of the Hemispheric Transport of Air Pollution (HTAP1). They found that 31% of
South Asia sulfate aerosol optical depth over South Asia was contributed by non-local
sources. Based on the HTAP2, Stjern et al. (2016), using results from ten models,
further assessed global and regional DRF from a 20% reduction in emissions over
seven regions including North America, Europe, South Asia, East Asia, Russia, the
Middle East, and the Arctic. They found that the 20% reduction in emissions in South
Asia and East Asia largely perturbed the radiative balance for other regions. However,
these studies focused on only the limited number of source regions over the Northern
Hemisphere. Continents and subcontinents over the tropics and Southern
Hemisphere are also important source and receptor regions for the sulfate radiative
forcing, especially for indirect forcing due to stronger aerosol-cloud interactions in
clean environments (Koren et al., 2014). Bellouin et al. (2016) quantified the radiative
forcing efficiency based on simulations of a 20% reduction in emissions from four
source regions/sectors in year 2008, and reported that, with aerosol-cloud
interactions included, models simulated higher radiative forcing efficiency of sulfate
compared to previous studies (Myhre et al., 2013, Shindell et al., 2013; Yu et al.,
2013). Few studies have quantified systematically the global source-receptor





relationships of sulfate indirect radiative forcing that can be attributed to
local/non-local source regions and anthropogenic/natural source sectors.

In this study, we introduce an explicit sulfur tagging technique into the

Community Earth System Model (CESM), in which sulfate aerosol and its precursor
emissions from fourteen major source regions and two natural source sectors are
tagged and explicitly tracked. We quantify source region/sector contributions to
regional and global sulfate mass concentrations, and direct and indirect radiative
forcing (DRF and IRF) of sulfate.

Model description, emissions datasets, and model experiments are shown in

Sect. 2. Section 3 gives the comparison of modeled concentrations of sulfate and
$SO_2$ with a variety of observations. Section 4 shows model results for source
attributions of near-surface sulfate and $SO_2$ concentrations over various receptor
regions. Source attributions of DRF and IRF of sulfate are discussed in Section 5.
Section 6 summarizes all the results and main conclusions.

**2. Methods**

We use the version 5 of the Community Atmosphere Model (CAM5), which is the

atmospheric component of CESM (Hurrell et al., 2013), to simulate the sulfate aerosol
and calculate its DRF and IRF. The modal aerosol treatment in CAM5 (Liu et al., 2012)
predicts number mixing ratios and mass mixing ratios of aerosols, distributed in three
lognormal modes. A set of modifications to CAM5 that improves wet scavenging of
aerosols and convective transport reported by Wang et al. (2013) has also been



implemented in the model used in this study. Parameterizations of aerosol optical
properties, cloud droplet nucleation, and aerosol-cloud interactions are described in
Neale et al. (2012). In addition to the standard radiative fluxes calculated with all
aerosols included, the CESM model has the capability of diagnosing radiative fluxes
for a subset of aerosol species. The difference between the standard and the
diagnosed shortwave radiative fluxes represents the DRF of the excluded aerosol
components in the diagnostic calculation (Ghan, 2013). To investigate IRF of sulfate
from different sources, we define in this study an incremental IRF, calculated as the
difference of cloud radiative forcing by perturbing 20% of sulfate and its precursor
emissions. Note that, the model only considers aerosol effects on stratiform cloud
(Morrison and Gettelman, 2008), and no microphysical impact on convective clouds
is included in the present version.

To quantify the regional source attributions of sulfate, for the first time, we

implemented in CESM/CAM5 a sulfur source-tagging capability, similar to the black
carbon tagging method used in H. Wang et al. (2014) and Yang et al. (2017), through
which sulfur gases and sulfate aerosols produced by emissions from independent
sources are tagged. The tool can be used to quantify the source attributions of $SO_2$
and sulfate without perturbing source emissions. The black carbon tagging only
required tagging interstitial and cloud-borne black carbon in the accumulation mode.
In contrast, the sulfur tagging requires tagging of interstitial and cloud-borne sulfate in
each of the three modes as well as $SO_2$, $H_2SO_4$ and dimethyl sulfide (DMS) gases. In
this study, sulfur species produced by emissions from fourteen geographical source



regions and two natural source sectors including volcanic eruptions and DMS from
oceans are tagged. The tagged and untagged models have been verified of producing
the same $SO_2$/sulfate properties and meteorology. While emissions of organic carbon,
black carbon, sulfate and its precursor gases are all included in the simulations, the
source tagging is used for sulfate and its precursor gases emissions alone.

The CEDS (Community Emissions Data System) anthropogenic emissions

(Hoesly et al., 2017) and open biomass burning emissions from Van Marle et al. (2017)
that were produced for the CMIP6 model experiments are used in our simulations. In
CAM5, 97.5% of $SO_2$ is emitted directly into the atmosphere and 2.5% is emitted as
sulfate aerosol. Natural emissions of volcanic $SO_2$ and DMS are the same as those
used in AeroCom following Neale et al. (2012), which are kept constant throughout the
selected years in this study. Figure 1a shows the fourteen geographical source
regions tagged in this study, which are consistent with source-receptor regions
defined in HTAP2, including North America (NAM), Central America (CAM), South
America (SAM), Europe (EUR), North Africa (NAF), Southern Africa (SAF), the
Middle East (MDE), Southeast Asia (SEA), Central Asia (CAS), South Asia (SAS),
East Asia (EAS), Russia/Belarus/Ukraine (RBU), Pacific/Australia/New Zealand
(PAN), and rest of the world (ROW, including oceans and polar continents). Table 1
summarizes emissions of combustion $SO_2$ (anthropogenic + open biomass burning),
volcanic $SO_2$ emissions (VOL), and DMS emissions over the sixteen tagged source
regions/sectors averaged for the most recent five years (2010–2014) and Figure 1b
presents relative contributions from individual source regions to the global





combustion $SO_2$ emissions. The global combustion $SO_2$ emissions rate is 57.6 Tg S
$yr^{-1}$, of which more than 98% come from anthropogenic sources. The combustion $SO_2$
and sulfate are referred to anthropogenic $SO_2$ and sulfate hereafter. Detailed
information on the anthropogenic emissions of $SO_2$ can be found in Hoesly et al.
(2017). East Asia, with regional emission of 17.8 Tg S $yr^{-1}$ (31% of global
anthropogenic $SO_2$), has the largest total $SO_2$ emissions, compared to the other
tagged regions. South Asia also emits a large amount of $SO_2$, 6.4 Tg S $yr^{-1}$ (11%),
followed by 3.4 Tg S $yr^{-1}$ (6%) from the Middle East, 3.3 Tg S $yr^{-1}$ (6%) from Europe,
3.1 Tg S $yr^{-1}$ (5%) from North America, and 2.7 Tg S $yr^{-1}$ (5%) from Southern Africa.
The other individual tagged regions have weaker emissions, with a combined
contribution of less than 5%. However, emissions from ROW contribute 11.2 Tg S $yr^{-1}$
(19%) of $SO_2$ that are mainly from shipping emissions near the continents. In addition,
natural emissions of sulfur are also accounted for, including 12.6 Tg S $yr^{-1}$ of $SO_2$ from
volcanic eruptions, in the range of 10–13 Tg S $yr^{-1}$ derived from the Ozone Monitoring
Instrument (OMI) measurement (McLinden et al., 2016), and 18.2 Tg S $yr^{-1}$ of DMS.
Figure 2 shows the spatial distribution of $SO_2$ emissions from each tagged
region/sector as well as DMS emissions. Emissions are spatially heterogeneous even
within the individual tagged regions. For instance, $SO_2$ emissions in North America are
mainly located in Eastern U.S., and Eastern China accounts for the majority of $SO_2$
emissions from East Asia. In addition, seasonal variations in emissions are quite
different among the source regions (Table 1). East Asia, RBU and Europe have
seasonal peak emissions in boreal winter, and Southern Africa shows larger



emissions in boreal summer, while emissions from North America are comparable in
winter and summer. Although volcanic $SO_2$ emissions are scattered near continents, a
large amount of them are injected into the free troposphere. DMS is emitted over
oceans with a boreal winter peak. These heterogeneous spatial and temporal
distributions of emissions could lead to different influences on air quality and radiative
forcing over continents and subcontinents near the source regions.

The CAM5 simulation is conducted using a meteorological nudging method (Ma

et al., 2013; Zhang et al., 2014), with winds nudged to the MERRA reanalysis
(Rienecker et al., 2011) every 6 hours. The simulation is integrated for years 2009–
2014, with 2009 for spin-up and 2010–2014 for analysis. A sensitivity simulation with
the same model configuration but having a uniform 20% reduction in sulfur ($SO_2$,
sulfate, DMS) emissions globally is performed to quantify source attributions of
incremental IRF of sulfate. Two additional sensitivity simulations with the same
standard model configuration but having a 20% reduction in global DMS emissions
and regional sulfur emissions over North America, respectively, are performed to
validate the decomposition of global incremental IRF into contributions from source
regions/sectors using the tagging method. And one additional sensitivity simulation
with anthropogenic $SO_2$ emissions fixed at 1850 level globally is performed to
compare incremental IRF and anthropogenic IRF of sulfate. All simulations are
performed at 1.9° latitude by 2.5° longitude horizontal grids and 30 vertical layers.
**3. Model evaluation**



To evaluate the model's performance in simulating sulfate with the latest
emissions from CEDS inventory, the simulated sulfur concentrations are compared
with measurements from regional observation networks. These datasets include the
Interagency Monitoring of Protected Visual Environments (IMPROVE), the European
Monitoring and Evaluation Programme (EMEP), the East Asian Monitoring Network
(EANET), and the China Meteorological Administration Atmosphere Watch Network
(CAWNET, Zhang et al., 2012). Sulfate concentrations observed from IMPROVE,
EMEP and EANET being used here are from 2010 to 2014, covering the same time
period as the simulation, while CAWNET only collected data over 2006–2007. In
order to use the CAWNET data to evaluate 2010-2014 simulation results, we decide
to scale the observed sulfate mass concentrations using the ratio of CEDS
2010-2014 $SO_2$ emissions to 2006-2007 emissions over China (which is 0.92) for
comparison, thus assuming a linear relationship between $SO_2$ emissions and sulfate
concentrations.
Figure 3 shows the comparison of modeled annual mean near-surface sulfate
concentrations with those from the observational networks. The model successfully
reproduces the global spatial distribution of sulfate with high concentrations over East
Asia and low concentrations over North America and Europe, as well as the spatial
patterns within major continents, for instance, high (low) values over Eastern
(Western) U.S. and high (low) sulfate concentrations over Eastern (Western) China.
The spatial correlation coefficient between simulated and observed sulfate
concentrations globally is +0.86 and is statistically significant at the 95th percentile.





Compared to the measurements at the IMPROVE sites over North America, at the
EMEP sites over Europe, and at the EANET sites over part of East Asia (only one site
in China) and Southeast Asia, the model reproduces sulfate concentrations with
biases within ±20%. However, the model largely underestimates the simulated sulfate
concentrations in China, with normalized mean biases (NMB) of −54%, compared to
the CAWNET observations.
A few factors could be responsible for the bias between the observed and
modeled sulfate concentrations. Underestimation of local $SO_2$ emissions could result
in the simulated low sulfate concentrations (Liu et al., 2012; Wang et al., 2013). Too
frequent liquid clouds and too strong wet scavenging at the mid- and high latitudes in
CESM model can lead to shorter aerosol lifetime and lower concentrations in the
simulation (Wang et al., 2011; Liu et al., 2012; Wang et al., 2013). In addition, the
underestimation of emissions from upwind regions or strong wet scavenging of
aerosols during transport could be another reason for the simulated low bias (Yang et
al., 2017). A too low rate of transformation from $SO_2$ gas to sulfate particles in the
model could also contribute to the low bias in sulfate concentrations (Wang et al.,
2016; Li et al., 2017). The bias can also result from the fact that the site
measurements are point observations, while the model results are grid-cell average
that does not consider subgrid aerosol variations (Qian et al., 2010; R. Wang et al.,
2014). Considering the longer lifetime of $SO_2$/sulfate than black carbon, this effect
would be expected to be less significant for $SO_2$/sulfate. In addition, different models
show large discrepancies in simulating sulfate over China (Kasoar et al., 2016). The





underestimation of sulfate in China can lead to an underestimation of source
contribution from East Asia of sulfate concentrations, direct and indirect radiative
forcing of sulfate, and forcing efficiencies of sulfate.

To evaluate the model results more broadly, we compare the simulated total

column burden of $SO_2$ with that derived from the OMI measurements (Li et al., 2013),
as shown in Fig. S1. Both the model results and the OMI satellite data are averaged
over 2010–2014. Compared to the OMI $SO_2$, the spatial distribution of column burden
of $SO_2$ is reproduced in CAM5, with a statistically significant spatial correlation
coefficient of +0.57. However, the model largely overestimates the magnitude of $SO_2$,
especially over China where the simulated values are about 8 times larger than OMI
data. The large difference between $SO_2$ burden and OMI retrievals over China must
be due to either an underestimation of $SO_2$ in OMI products and/or an overestimation
of $SO_2$ burden in the model results. He et al. (2012) compared in situ measurements
with OMI $SO_2$ burden over central China and reported a negative bias of 50% in OMI
data, which probably came from cloud contamination, reduced satellite sensitivity to
$SO_2$ due to aerosols, and spatial sampling bias in the satellite data. It is also worth
mentioning that satellite column-$SO_2$ retrievals depend on the vertical distribution of
$SO_2$ assumed in the retrieval algorithm, which could be different from either the
modeled $SO_2$ profile in this study or the actual profile, which would introduce a bias.

The simulated $SO_2$ near-surface concentrations, however, are also

underestimated by 25% compared to observations over thirteen sites in China (Gong
et al., 2014) shown in Fig. S2a, also suggesting a large bias in satellite retrievals or too





much $SO_2$ simulated in higher altitude. The modeled $SO_2$ concentrations over
downwind regions of China are underestimated by 45% compared to observations
from EANET sites (Fig. S2b), indicating that the transport of $SO_2$ from China is
probably underestimated in the model.

A less efficient of transformation from $SO_2$ to sulfate could also lead to

underestimation of sulfate. A recent study by Wang et al. (2016) focusing on the
sulfate pollution over China and London found that aqueous oxidation of $SO_2$ by $NO_2$
was key to an efficient sulfate formation, which has typically been neglected in
atmospheric models and is not considered in the CAM5. Another study by Li et al.
(2017) found that including an aerosol water ($HRSO_2$) parameterization in $SO_2$
oxidation in a box model could reproduce the observed rapid sulfate formation in Xi'an
over China. More rapid oxidation of $SO_2$ would reduce $SO_2$ loss by dry and wet
removal and increase sulfate production, which can partly explain the low bias in the
simulated sulfate concentrations and high bias in $SO_2$. In CAM5, 36% of total sulfur
converts into column-integrated sulfate over China, similar to 33% in the Community
Multiscale Air Quality (CMAQ) model (He et al., 2012). However, it changes to 21% in
the bottom model layer (about 992 hPa), indicating that the oxidation of $SO_2$ may be
underestimated near the surface, which most directly affects the comparison to
near-surface observations. This appears to be a plausible explanation for the
underestimated sulfate concentrations over China and points to a potentially important
direction for future model development.
**4. Source attribution of sulfate mass concentrations**





Figure 4 shows spatial distributions of modeled fractional contributions to annual
near-surface sulfate concentrations. (The absolute concentrations of sulfate are
shown in Fig. S3). East Asia, ROW, South Asia and the Middle East contribute 16%,
14%, 10% and 7%, respectively, to global annual mean near-surface sulfate
concentration, whereas contributions from the other individual source regions are all
less than 5%. Natural emissions of volcanic $SO_2$ and ocean DMS account for 11% and
16% of global mean sulfate concentrations. Sulfate concentrations are mostly
contributed by local sources in regions with high emissions, such as Eastern U.S.,
Southern Africa, South Asia, and Eastern China, where local source contributions are
larger than 80%. Over regions with relatively low $SO_2$ emissions, the near-surface
sulfate concentrations are primarily attributed to non-local sources from long-range
transport. Natural DMS emissions are the source of 80% of near-surface sulfate
concentrations over Southern Hemisphere oceans and 20–60% for Northern
Hemisphere oceans. Over downwind ocean regions of East Asia, emissions from
DMS only account for 20–40% of near-surface sulfate concentrations, showing a
stronger influence of regional transport. Sources from volcanic eruption strongly
influence sulfate concentrations over eruption regions. They are responsible for 10–
40% of near-surface concentrations over Central America and South America, 40–80%
over North Africa and Southeast Asia, but only account for about less than 5% over
East Asia and South Asia where anthropogenic emissions dominate.
The spatial distribution of sulfate column burden and relative contributions are
shown in Figs. S4 and S5, respectively. The global average source attribution of





column burden does not differ significantly from that of near-surface concentration.
The exception is an increase from 11% to 15% of the relative contribution from VOL to
column burden as compared to near-surface concentration due to injection mostly into
the free troposphere. The DMS contribution decreases from 16% to 11% to
compensate the increase of VOL contribution over oceans. In general, the relative
contribution from local source to column burden within a source region is lower than
that of near-surface concentration.

Figure 5 presents relative contributions of major sources to near-surface sulfate

concentrations in neighboring receptor regions along with seasonal mean wind fields
at 850 hPa. (Table S1 summarizes a complete list of numbers characterizing the
source-receptor relationships.) Transport of sulfate shows different patterns in
different seasons, due to the seasonal variability in local precursor emissions, lifetime
of sulfate, and meteorology, such as wind fields and precipitation. Sulfate originating
from North America, Central America and South America do not show significant
contributions (relative contribution less than 10%) to sulfate over other tagged regions
in all seasons because of the relatively low sulfate concentrations over these regions
and the long intercontinental transport pathways.

The export of sulfate from Europe contributes to about 16–20% of near-surface

sulfate concentrations over North Africa, RBU and Central Asia in all seasons due to
the westerly jet over the eastern European boundary and northerly winds over
southern boundary. Sulfate concentrations in North Africa and Southern Africa are
relatively low and there is no significant export to other regions.

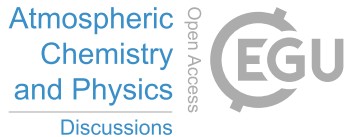

The Middle East has relatively high concentrations of sulfate (Fig. S3). Sulfate
from this region can be effectively transported to the surrounding receptor regions:
North Africa, Southern Africa, Central Asia, and South Asia. This export accounts for
15–24% of sulfate concentrations over North Africa, Southern Africa and Central Asia
in DJF and SON, and 19% over South Asia in MAM. Sources in the RBU explain
about 21–42% of sulfate concentrations over Central Asia, especially in JJA, with
northerly winds over north boundary of Central Asia driving transport from this region.
Central Asia does not have a discernable export of sulfate due to low emissions in
most seasons, except that it accounts for 13% of sulfate over the RBU region in DJF
when source emissions are the largest.
Northerly winds over East Asia in DJF and SON associated with the East Asian
winter monsoon transport sulfate from highly polluted Eastern China to Southeast
Asia, which accounts for about 50% of near-surface sulfate concentrations over
Southeast Asia in these months. The oxidation of $SO_2$ is expected to peak in JJA
because of the high temperature and humidity, and more sunlight. With the help of
southerly winds of East Asian summer monsoon, East Asia contributes to 15% of
sulfate concentrations over RBU in JJA. Due to the strong westerly jet in MAM,
sulfate originating from East Asia has a long-range transport across the North Pacific
and accounts for 11% of near-surface sulfate concentrations and 25% of total
imported sulfate (without local contributions) over North America. The transport of
sulfate from South Asia contributes 11–24% of sulfate in Southeast Asia in DJF and




MAM. These results, however, have additional uncertainties due to the $SO_2$/sulfate
bias in the model for East Asia discussed previously.

Source-receptor relationships for sulfate column burden are summarized in Table

S2. Compared to the near-surface concentrations, the sulfate column burden
contributed by local sources is much lower in all the receptor regions due to the more
efficient long-range transport of aerosols in the free atmosphere. Annually, the local
contribution over North America decreases from 67% for near-surface concentration
to 33% for column burden. The contributions of non-local sources from East Asia and
South Asia increase from 7% and 1% for near-surface concentration to 24% and 10%
for column burden, respectively, to the sulfate over North America. In addition, South
Asia contribution to sulfate in East Asia, and East Asia contribution to sulfate in RBU
and Europe also significantly increase for column burden compared to near-surface
concentrations.

Figure 6 shows local contributions (i.e., from sources within the tagged regions)

to near-surface sulfate concentrations. Averaged over individual tagged regions,
contributions from local sources dominate (i.e., local contributions > 50%) over North
America, South America, Europe, Southern Africa, the Middle East, South Asia, and
East Asia. Imports dominate near-surface sulfate concentrations (i.e., local
contributions < 50%) over the rest of tagged land regions. Within each tagged region,
whether local source or import dominates depends on specific locations. For instance,
over Eastern China, because of high anthropogenic emissions, local contribution to
sulfate concentration is larger than 80%, whereas import from other source regions


dominates sulfate over the less economically developed Western China. The same
difference can be found between Eastern and Western U.S. of the tagged North
America. Over oceans in the Southern Hemisphere, natural sources of DMS
contribute the largest to local sulfate concentrations (Fig. 4), whereas long-range
transport dominates over the North Pacific in DJF and MAM.

Figure 7 presents the aggregate, seasonal relative source contributions to area

weighted average near-surface sulfate concentrations over land/ocean in the
Northern/Southern Hemisphere. Over land in the Northern Hemisphere, sulfate
concentration is mainly attributed to sources from East Asia, South Asia, the Middle
East, ROW and volcanic eruption, with relative contributions of 22–29%, 9–16%, 8–
14%, 9–11%, and 6–13%, respectively. Over ocean in the Northern Hemisphere,
although contribution from ROW, volcanic $SO_2$ and DMS increase dramatically
compared to land, contributions from East Asia and South Asia do not have a large
decrease, especially in DJF, MAM and SON when aerosol outflow from Asia is strong
(Yu et al., 2012; Yang et al., 2015). Over land in the Southern Hemisphere, mean
sulfate concentration is dominated by sources in Southern Africa, having a
contribution of 33–43%, followed by 13–25% from South America. Emissions from
DMS drive sulfate over ocean in the Southern Hemisphere in all seasons contributing
27–63% of sulfate, although Southern Africa contributes 20% of sulfate in JJA.

Figure 8 shows seasonal and annual mean regional concentration efficiencies of

sulfate from the tagged source regions/sectors, defined as the local contribution to
near-surface sulfate concentration divided by the corresponding sulfur emissions



from that region. (Table S3 provides the numeric values.) The regional concentration
efficiency represents the relationship between local contribution to sulfate
concentration and local emission, which is influenced by many factors, such as local
production of sulfate from the emitted $SO_2$, aerosol removal and export. Note that, the
receptor region of ROW is used to calculate efficiencies of the VOL and DMS source
sectors, which leads to low biases in efficiencies. The efficiencies over the Middle
East show high values in almost all seasons due to dry atmospheric conditions
favoring long aerosol lifetime (e.g., Wang et al., 2014; Stjern et al., 2016). The
efficiencies are also high over South Asia in DJF and SON, but low in MAM and JJA
due to strong wet removal during the South Asian summer monsoon season. North
Africa and Central Asia also show high efficiencies resulted from less precipitation.
Although East Asia does not have much precipitation in DJF, the efficiency is low
because a large amount of sulfate is transported outside East Asia. It suggests that
the lifetime of aerosols, mainly driven by wet deposition, together with regional export,
is important in determining the local contribution to near-surface concentrations or
regional air quality.

**5. Source attribution of direct and indirect radiative forcing of sulfate**
The modeled global annual mean sulfate total DRF here is $-0.42$ W m$^{-2}$, with $-$
0.31 W m$^{-2}$ contributed by anthropogenic sulfate and $-0.11$ W m$^{-2}$ contributed by
natural sulfate (e.g., relative to a state with no natural emissions). The DRF of
anthropogenic sulfate is $-0.4\pm0.2$ W m$^{-2}$ provided in the Fifth Assessment Report of





the Intergovernmental Panel on Climate Change (IPCC, 2013). Note that, the DRF of
anthropogenic sulfate calculated here is total anthropogenic sulfate, whereas values
from IPCC represent changes in anthropogenic sulfate between 1750 and
present-day conditions, although this difference is small since 1750 $SO_2$ emissions
are less than 1% of 2010 emissions. Spatial distributions of sulfate DRF, originating
from the individual sixteen sources are shown in Fig. S6. The spatial distributions and
global contributions of sulfate DRF are similar to those of sulfate column burden (Fig.
S4), except that contribution of DMS to global sulfate DRF (18%) is much larger
relative to its global column burden (11%). It is because DMS-produced sulfate
burden is mostly located between 30°S–30°N (Fig. S4), where insolation is much
stronger than at mid- and high latitudes, leading to stronger DRF over these regions.
East Asia is the second largest contributor to global sulfate DRF, contributing 16% of
global sulfate DRF, followed by 13% from ROW and 11% from South Asia.

Figure 9 shows seasonal and zonal mean DRF of sulfate originating from the

tagged regions/sectors and the global total. The meridional distribution of DRF is
jointly determined by many factors, e.g. sulfate loading, the insolation, cloud cover,
and surface albedo. The total sulfate DRF shows a seasonal pattern that has the
maximum DRF over 0°–10°N in DJF and over 30°–40°N in JJA, with values between
–0.9 and –1.3 W m$^{-2}$. Emissions originating from East Asia have the largest zonal
mean sulfate DRF in almost all seasons, with a maximum around –0.45 W m$^{-2}$ in JJA
because of the higher sulfate loading and the more abundant sunlight in JJA. South
Asia also strongly contributes to sulfate DRF, followed by sources from ROW, VOL,



RBU, and the Middle East. DMS has the largest contribution over 60°S–0° in DJF due
to the stronger insolation over the Southern Hemisphere in winter and more ocean
DMS emission over these regions, and its DRF contribution is more widespread.
Other tagged source regions have a relatively small contribution to the global total
DRF, with a seasonal peak DRF less than –0.10 W m$^{-2}$. The global and annual
average sulfate DRF has a contribution of –0.074 W m$^{-2}$ from DMS, –0.068 W m$^{-2}$
from East Asia, –0.054 W m$^{-2}$ from ROW, –0.047 W m$^{-2}$ from South Asia, –0.035 W
m$^{-2}$ from VOL, –0.031 W m$^{-2}$ from the Middle East, –0.023 W m$^{-2}$ from Southern Africa,
–0.018 W m$^{-2}$ from Europe, –0.016 W m$^{-2}$ from North America, and a total of –0.057
W m$^{-2}$ from all other regions (Table S4).

Figure 10 shows seasonal fractional contributions to sulfate DRF in different

latitudinal bands. Over the Southern Hemisphere tropics (30°S–Equator), mid-
(60°S–30°S) and high (90°S–60°S) latitudes, DMS has the largest contribution to
sulfate DRF in all seasons, with contribution about 17–84%. Sources from Southern
Africa contribute about 11–20% of sulfate DRF over the Southern Hemisphere tropic
and mid-latitudes, followed by about 10% from South America and ROW. Sources
from East Asia account for 6–19% of sulfate DRF over the Southern Hemisphere high
latitudes. In the Northern Hemisphere, influence from DMS becomes much weaker,
but still substantial. Over the Northern Hemisphere tropics, East Asia, South Asia,
ROW, and DMS exert equal contributions of 10–20%. East Asia has the largest
contribution of 20–30% over the Northern Hemisphere mid- and high-latitudes,
followed by South Asia and ROW.



Sulfate incremental IRF is estimated by using an additional simulation in which
sulfur emissions are reduced by 20% for all regions and sectors. The difference in
cloud radiative forcing between the control simulation and this second simulation gives
the sulfate incremental IRF of the last 20% of sulfur emissions. Regional incremental
IRF contributions are calculated by scaling the total incremental IRF in a grid column
by regional source contributions to sulfate mass concentration reduction averaged
from the surface layer to 850 hPa, which is the approximate altitude of cloud base.
Figure 11 shows regional contributions to sulfate incremental IRF from the tagged
source regions/sectors. The sulfate incremental IRF is $-0.44$ W m$^{-2}$. The spatial
pattern is consistent with that of stratiform clouds since the model only considers
aerosol effects on stratiform cloud. The strong negative forcing is mainly over oceans.
All source contributions to sulfate incremental IRF from the fourteen tagged source
regions are less than $-0.04$ W m$^{-2}$, probably due to the polluted conditions over or near
land. Particles originating from North America, South America, Southern Africa, and
East Asia are also transported to ocean regions, leading to a strong negative forcing
there. DMS has the largest contribution, explaining $-0.23$ W m$^{-2}$ of the global sulfate
incremental IRF, because complex cloud adjustments are likely to respond sensitively
to small changes in aerosol under clean conditions (Rosenfeld et al., 2014), followed
by $-0.06$ W m$^{-2}$ from volcanic emissions. Note that the regional contribution to
incremental IRF is simply calculated by decomposing the total incremental IRF with
mass concentrations based on two simulations without and with the reduction in



emissions. This assumption could introduce biases considering non-linear relationship
between mass concentration and IRF of sulfate.

To evaluate this new method for decomposing incremental IRF into different

source regions/sector contributions, the IRF for one region (North America) and one
sector (DMS) were calculated in a traditional manner using two additional simulations
in which $SO_2$ emissions from North America and DMS emissions were reduced by
20%, respectively. The incremental IRF calculated with the two methods are
compared in Fig. S7. Although the incremental IRF outside the source regions
obtained from the emission perturbation method is noisy, these two methods show
similar negative incremental IRF within and near source regions. The 20% DMS leads
to strong negative IRF over oceans and sources from North America result in negative
IRF over Eastern U.S. and downwind ocean regions for both of these two methods.
Globally, DMS and North America contribute to –0.231 (±0.012) and –0.014 (±0.002)
W m$^{-2}$, respectively, of sulfate incremental IRF from the method with sulfur tagging
technique, similar to –0.248 (±0.020) and –0.018 (±0.019) W m$^{-2}$ from the individual
emission-perturbation simulations.

Table S5 summarizes the DRF and incremental IRF of sulfate over land/ocean in

the Northern/Southern Hemisphere contributed by the tagged source regions/sectors.
Over the fourteen tagged source regions, the total anthropogenic source region
contribution to DRF is –0.54/–0.18 W m$^{-2}$ over land in the Northern/Southern
Hemisphere, larger than –0.48/–0.12 W m$^{-2}$ over ocean due to the larger sulfate
burden near sources. Anthropogenic source contributions to incremental IRF are





larger over ocean, with values of –0.23/–0.13 W m$^{-2}$ compared to –0.08/–0.10 W m$^{-2}$
over land in the Northern/Southern Hemisphere, because clouds are more
susceptible to aerosol changes in clean environment and there are more stratiform
clouds over ocean. For natural source sectors, their contributions are larger over
oceans for both DRF and incremental IRF. Over land in the Northern Hemisphere,
DRF is mainly driven by emissions from East Asia, South Asia, and the Middle East,
whereas incremental IRF is dominated by emissions from North America, RBU and
East Asia. The difference in major contributing regions for DRF vs. incremental IRF
may be due to changes in cloud susceptibility when background aerosol
concentrations are different. North America and RBU have more relatively clean
areas (Alaska, N. Canada, parts of Siberia) than South Asia and East Asia, and
clouds in the cleaner areas are more susceptible to the 20% emissions reductions.
The non-linearity in DRF is much weaker, so the high emissions from South Asia and
East Asia dominate DRF. Over ocean in the Northern Hemisphere, East Asia also
contributes the largest to DRF and it is the second largest contributor to incremental
IRF of sulfate following DMS. Over land in the Southern Hemisphere, emissions from
Southern Africa and South America control DRF, whereas incremental IRF are
largely attributed to sources from South America, DMS, and PAN
(Pacific/Australia/New Zealand). Over ocean in the Southern Hemisphere, both
sulfate DRF and incremental IRF are dominated by DMS emissions.

Figure 12 shows the seasonal and annual global DRF and incremental IRF

efficiencies of sulfate. (Table S6 gives values.) Global DRF efficiency of a source


region is defined as the global DRF of sulfate originating from the source
region/sector divided by the total sulfur emissions from that region/sector. The global
DRF efficiency treats the whole globe as a receptor region, as opposed to a specific
region in the regional concentration efficiency definition, considering that aerosol
climatic impacts are on a global scale whereas air quality impacts are more important
on a regional scale. As the DRF is more closely related to sulfate burden, global
sulfate burden efficiencies are also provided in Table S7. The global DRF efficiency
for total sulfur emissions is $-4.8$ mW m$^{-2}$ (Tg S yr$^{-1}$)$^{-1}$. The Middle East, North Africa,
and Southern Africa present high DRF efficiencies, as a result of both long aerosol
lifetime and strong tropical insolation. These source regions also have high global
burden efficiencies.
Table S6 compares the annual DRF efficiencies simulated in this study with the
average from a previous multi-model study (Stjern et al., 2016). In Stjern et al. (2016),
efficiency was calculated as the response of global DRF to a 20% reduction in local
emissions divided by the 20% of total SO$_2$ emissions based on two separate
simulations, whereas the efficiency in this present study is calculated as the global
contribution of DRF divided by 100% of local emissions in a single simulation, taking
advantage of the sulfur tagging capability and radiation diagnostic calculations. Both
studies show high efficiencies over the Middle East and South Asia, and low
efficiencies over North America, Europe, East Asia and RBU. In addition, the
magnitude of efficiencies in this study are very similar to Stjern et al. (2016), with
differences less than 15%.





The global IRF efficiency of a source region is calculated as the global
contribution of sulfate incremental IRF divided by the changes (i.e., 20% reduction) in
sulfur emissions in that region. Unlike the DRF efficiencies, IRF efficiencies are
higher over or near ocean regions, with a global IRF efficiency of $-5.0$ mW m$^{-2}$ (Tg S
yr$^{-1}$)$^{-1}$ for the global total 20% of sulfur emissions. PAN and DMS have the largest IRF
efficiencies because PAN has a relatively clean environment compared to other
regions and DMS is emitted over clean oceans. Cloud properties are more
susceptible to aerosol perturbations in a more pristine environment. Although the
background aerosols in South America are not so low, sulfate originating from this
region has a large contribution to sulfate over oceans of the Southern Hemisphere,
explaining a large IRF efficiency from that region.
In addition to the incremental IRF and efficiency, we also calculated the
anthropogenic sulfate IRF and its efficiency between present-day and preindustrial
conditions with an additional simulation, in which anthropogenic $SO_2$ emissions are
fixed at the 1850 level, and compared these values with those from the 20% sulfur
emission reduction simulation in Table S8. The modeled annual and global mean
anthropogenic sulfate IRF here is $-0.74$ W m$^{-2}$, which is comparable to $-0.45 \pm 0.5$ W
m$^{-2}$ of IRF for total anthropogenic aerosols from IPCC (2013). The anthropogenic IRF
contributed from individual source regions is about 3–6 times larger than the
incremental IRF, in agreement with about 5 times more reduction in $SO_2$ emissions in
the preindustrial simulation than in the 20% sulfur emission reduction simulation. The
forcing efficiencies are roughly similar between the incremental and the



anthropogenic IRF, indicating a nearly linear relationship between $SO_2$ emission and
sulfate IRF, except for the Middle East and South Asia where concentrated dust and
its variability may strongly influence cloud properties and therefore sulfate IRF. Figure
S8 shows the anthropogenic sulfate IRF efficiencies that are calculated based on
anthropogenic IRF from the present-day and preindustrial condition simulations. The
values are similar to the incremental IRF efficiencies, further validating the robust
results from the decomposed regional IRF with the sulfur tagging technique.

**6. Conclusions and discussions**
A sulfur tagging technique is implemented in Community Atmosphere Model
(CAM) of the Community Earth System Model (CESM) and used in this study to
examine source-receptor relationships of sulfate concentrations, DRF and IRF
originating from sixteen regions/sectors (North America, Central America, South
America, Europe, North Africa, Southern Africa, the Middle East, Southeast Asia,
Central Asia, South Asia, East Asia, RBU, PAN, ROW, VOL, and DMS) for 2010–
2014. The anthropogenic emissions came from the CEDS inventory developed for
the CMIP6.
Near-surface sulfate concentrations are mostly contributed by local emissions in
regions with high emissions, such as Eastern U.S., Southern Africa, South Asia, and
Eastern China, where local source contributions exceed 80%. Over regions with
relatively low $SO_2$ emissions, the near-surface sulfate concentrations are primarily
attributed to non-local sources from long-range transport.



The source-receptor relationships have strong seasonal variations. The export of
sulfate from Europe contributes to 16–20% of near-surface sulfate concentrations
over North Africa, RBU and Central Asia in all seasons. Sulfate from the Middle East
is effectively transported to the surrounding receptor regions and accounts for 15–24%
of sulfate concentrations over North Africa, Southern Africa and Central Asia in DJF
and SON, and 19% over South Asia in MAM. Sources in RBU account for 21–42% of
sulfate concentrations over Central Asia, with a peak contribution in JJA. Northerly
winds over East Asia in DJF and SON associated with East Asian winter monsoon
transport sulfate from highly polluted Eastern China to Southeast Asia, accounting for
about 50% of near-surface sulfate concentrations over Southeast Asia. East Asia
also contributes 15% to the near-surface sulfate over RBU in JJA and 11% over North
America in MAM. The transport of sulfate from South Asia contributes 11–24% of
near-surface sulfate over Southeast Asia in DJF and MAM. Regional sulfate
concentration efficiencies are higher over regions with dry atmospheric conditions
and less export, suggesting that the lifetime of aerosols mainly driven by wet
deposition, together with regional export, is important in determining the regional air
quality.
The simulated global total sulfate DRF is –0.42 W m$^{-2}$, with –0.31 W m$^{-2}$
contributed by anthropogenic sulfate and –0.11 W m$^{-2}$ contributed by natural sulfate.
DMS has the largest contribution to the global sulfate DRF, followed by East Asia,
ROW and South Asia. In the Southern Hemisphere, DMS contributes 17–84% to the
seasonal total sulfate DRF. In the Northern Hemisphere tropics, East Asia, South



Asia, ROW, and DMS exert similar contributions of 10–20%. East Asia has the
largest contribution of 20–30% over the Northern Hemisphere mid- and high-latitudes,
followed by South Asia and ROW.

Sulfate incremental IRF is estimated using an additional simulation in which sulfur

emissions are reduced by 20%. The difference in cloud radiative forcing between the
control simulation and this second simulation gives the sulfate incremental IRF of the
last 20% of sulfur emissions, which is –0.44 W m$^{-2}$ globally. DMS has the largest
contribution, explaining –0.23 W m$^{-2}$ of the global sulfate incremental IRF, because of
the clean marine background conditions, followed by –0.06 W m$^{-2}$ from volcanic
emissions.

The Middle East, North Africa, and Southern Africa have high global DRF

efficiencies, due to both longer aerosol lifetimes (from low precipitation) and strong
insolation. Regions in the Southern Hemisphere with low background aerosols have
stronger global IRF efficiencies than those over the polluted Northern Hemisphere,
because cloud properties are more susceptible to aerosol perturbations in a more
pristine environment.

Note that, although simulated near-surface sulfate concentrations are in

agreement with observed values at the IMPROVE sites over North America and at
the EANET sites over part of East Asia and Southeast Asia, the model strongly
underestimates sulfate concentrations by –54% in China, compared to site
observations from the CAWNET network. Comparison of column-integrated $SO_2$
between model simulation and OMI satellite data shows a possible overestimation of



$SO_2$ in the model. The simulated $SO_2$ near-surface concentrations, however, are
underestimated by 25% compared to observations over thirteen sites in China,
suggesting a large bias in satellite retrievals or too much $SO_2$ simulated at higher
altitudes. The model $SO_2$ concentrations over downwind regions of China are
underestimated by 45%, indicating that the transport of $SO_2$ from China is probably
underestimated in the model. A less efficient transformation from $SO_2$ to sulfate could
also lead to the underestimation of sulfate in the model. The underestimation of sulfate
over China could lead to the underestimation of contributions from East Asia to remote
sulfate concentrations, global DRF and incremental IRF, as well as their efficiencies.




*Data availability.* All the emissions datasets used in this study can be obtained from
https://pcmdi.llnl.gov/projects/input4mips. The sulfate datasets are available from
http://vista.cira.colostate.edu/IMPROVE/ for IMPROVE sites, http://www.eanet.asia
for EANET sites, and http://www.emep.int for EMEP sites. The OMI satellite-derived
total column burden of $SO_2$ can be downloaded from
http://disc.sci.gsfc.nasa.gov/Aura/data-holdings/OMI/omso2e_v003.shtml. The
CESM model is publically available at http://www.cesm.ucar.edu/models/cesm1.2/.
Our model results can be made available through the National Energy Research
Scientific Computing Center (NERSC) severs upon request.




*Competing interests.* The authors declare that they have no conflict of interest.

*Acknowledgments*. This research was supported by the National Atmospheric and
Space Administration's Atmospheric Composition: Modeling and Analysis Program
(ACMAP), award NNH15AZ64I. We also acknowledge support from the
U.S. Department of Energy (DOE), Office of Science, Biological and
Environmental Research. The Pacific Northwest National Laboratory is
operated for DOE by Battelle Memorial Institute under contract
DE-AC05-76RLO1830. The CESM project was supported by the National Science
Foundation and the DOE Office of Science. The National Energy Research Scientific
Computing Center (NERSC) provided computational resources.





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




**Table 1.** Seasonal emissions (units: Tg S season$^{-1}$) of combustion (anthropogenic +
biomass burning) $SO_2$ and DMS from the sixteen source regions/sectors in
December-January-February (DJF), March-April-May (MAM), June-July-August (JJA),
and September-October-November (SON) and annual total emissions (ANN).

|     | NAM | CAM | SAM | EUR | NAF | SAF | MDE | SEA |
|-----|-----|-----|-----|-----|-----|-----|-----|-----|
| DJF | 8.313E-01 | 3.458E-01 | 3.284E-01 | 1.073E+00 | 1.519E-01 | 6.507E-01 | 8.388E-01 | 3.537E-01 |
| MAM | 7.016E-01 | 3.659E-01 | 3.677E-01 | 8.251E-01 | 1.529E-01 | 5.871E-01 | 8.421E-01 | 3.731E-01 |
| JJA | 8.761E-01 | 3.731E-01 | 4.740E-01 | 6.456E-01 | 1.534E-01 | 8.090E-01 | 8.398E-01 | 3.516E-01 |
| SON | 7.045E-01 | 3.550E-01 | 4.357E-01 | 7.829E-01 | 1.518E-01 | 6.641E-01 | 8.353E-01 | 3.517E-01 |
| ANN | 3.114E+00 | 1.440E+00 | 1.606E+00 | 3.327E+00 | 6.099E-01 | 2.711E+00 | 3.356E+00 | 1.430E+00 |
|     | CAS | SAS | EAS | RBU | PAN | ROW | VOL | DMS |
| DJF | 3.156E-01 | 1.593E+00 | 5.043E+00 | 8.913E-01 | 1.266E-01 | 2.836E+00 | 3.106E+00 | 5.991E+00 |
| MAM | 2.720E-01 | 1.626E+00 | 4.406E+00 | 7.443E-01 | 1.352E-01 | 2.775E+00 | 3.175E+00 | 4.770E+00 |
| JJA | 2.300E-01 | 1.605E+00 | 4.084E+00 | 6.455E-01 | 1.597E-01 | 2.739E+00 | 3.175E+00 | 3.537E+00 |
| SON | 2.619E-01 | 1.594E+00 | 4.299E+00 | 6.940E-01 | 1.625E-01 | 2.813E+00 | 3.141E+00 | 3.918E+00 |
| ANN | 1.080E+00 | 6.418E+00 | 1.783E+01 | 2.975E+00 | 5.840E-01 | 1.116E+01 | 1.260E+01 | 1.822E+01 |





**Figure Captions**

**Figure 1.** (a) Tagged source regions (NAM: North America, CAM: Central America,

SAM: South America, EUR: Europe, NAF: North Africa, SAF: Southern Africa, MDE:

the Middle East, SEA: Southeast Asia, CAS: Central Asia, SAS: South Asia, EAS:

East Asia, RBU: Russia/Belarus/Ukraine, PAN: Pacific/Australia/New Zealand and

ROW: rest of the world) and (b) the respective percentage contributions to global

annual mean combustion $SO_2$ emissions (anthropogenic + biomass burning) from the

individual source regions.

**Figure 2.** Spatial distribution of annual mean emissions (g S m$^{-2}$ yr$^{-1}$) of

anthropogenic $SO_2$, volcanic $SO_2$, and DMS from the sixteen tagged source

regions/sectors averaged over 2010–2014.

**Figure 3.** Spatial distribution (left panel) and scatter plot (right) between the simulated

and observed annual mean near-surface sulfate concentrations (µg m$^{-3}$) over years

2010–2014. Observations are from IMPROVE (up pointing triangle), EMEP (square),

EANET (down pointing triangle) for years 2010–2014, and CAWNET (circle) for years

2006–2007, which are scaled to 2010–2014 based on the ratio of CEDS 2010-2014

$SO_2$ emissions to 2006-2007 emissions over China (which is 0.92). Solid lines mark

the 1:1 ratio and dashed lines mark the 1:2 and 2:1 ratio. Normalized mean bias

(NMB) and correlation coefficient (R) between observation and simulation are shown





on the right panel.  NMB = $100\% \times \sum (M_i - O_i) / \sum O_i$, where $M_i$ and $O_i$ are the
modeled and observed values at site $i$, respectively.

**Figure 4.** Spatial distribution of relative contributions (%) to annual mean
near-surface sulfate concentrations from each of the tagged source regions/sectors.
Relative contributions to global averaged sulfate from individual source
regions/sectors is shown at the bottom right of each panel.

**Figure 5.** Relative contributions of non-local sources to seasonal near-surface sulfate
concentrations (left panels) and wind fields over 850 hPa (right panels). Arrows with
numbers show contributions (%) of a source region to sulfate over a receptor region.
Only relative concentrations larger than 10% are shown.

**Figure 6.** Relative contributions (%) of local emissions (inside the tagged regions) to
near-surface sulfate concentrations. Contributions from natural source sectors are
added to ROW here. Contributions less than 50% are shown in cold colors and those
larger than 50% are shown in warm colors.

**Figure 7.** Relative contributions (%) to near-surface sulfate concentrations averaged
over land and ocean of the Northern and Southern Hemisphere from emissions in the
sixteen tagged source regions/sectors.



**Figure 8.** Seasonal and annual mean regional concentration efficiency of sulfate (µg
m$^{-3}$ (Tg S yr$^{-1}$)$^{-1}$) of the sixteen tagged source regions/sectors. The efficiency is
defined as the local contribution to near-surface sulfate concentration divided by the
corresponding sulfur emissions from that region (seasonal emissions multiplied by 4).
Error bars indicate 1-σ of mean values during years 2010–2014. The receptor region
of ROW is used to calculate efficiency of VOL and DMS.

**Figure 9.** Contributions to zonal mean sulfate direct radiative forcing (W m$^{-2}$) from
emissions of the tagged regions/sectors shown in colors (left Y axis) and from global
total emissions shown in black (right Y axis). Only regions with maximum of zonal
mean sulfate direct radiative forcing stronger than –0.1 W m$^{-2}$ are shown here.

**Figure 10.** Relative contributions (%) from emissions in the sixteen tagged
regions/sectors to sulfate direct radiative forcing over the Southern Hemisphere
high-latitudes (90°S–60°S), Southern Hemisphere mid-latitudes (60°S–30°S),
Southern Hemisphere tropics (30°S–Equator), Northern Hemisphere tropics
(Equator–30°N), Northern Hemisphere mid-latitudes (30°N –60°N), and Northern
Hemisphere high-latitudes (60°N –90°N).

**Figure 11.** Spatial distribution of responses of annual mean indirect radiative forcing
of sulfate (IRF, W m$^{-2}$) to a 20% reduction in sulfur emissions (standard simulation –
simulation with 20% emission reduction). Regional contributions are calculated as a



scaled total incremental IRF in each grid cell by the ratio of source contribution to total
sulfate mass concentration reduction averaged from the surface layer to 850 hPa.
Regional mean contributions to global incremental IRF of sulfate are shown at the
bottom right of each panel.

**Figure 12.** Seasonal and annual mean global sulfate (a) direct and (b) indirect
radiative forcing efficiency (mW m$^{-2}$ (Tg S yr$^{-1}$)$^{-1}$) of the sixteen tagged source
regions/sectors. The sulfate radiative efficiency is defined as the global sulfate
radiative forcing divided by the corresponding scaled annual sulfur emission
(seasonal emission multiplied by 4). Error bars indicate 1-σ of mean values during
years 2010–2014.




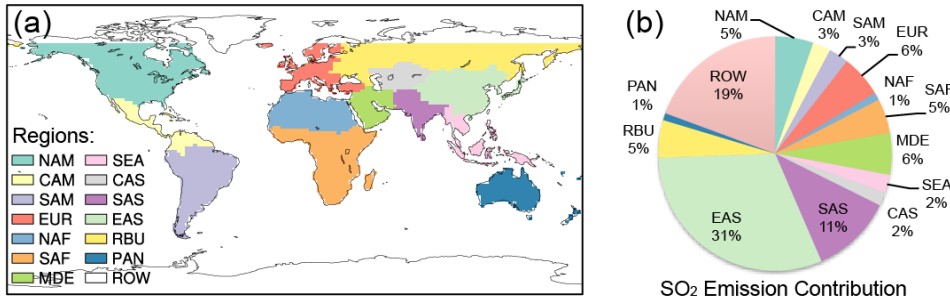



**Figure 1.** (a) Tagged source regions (NAM: North America, CAM: Central America, SAM: South America, EUR: Europe, NAF: North Africa, SAF: Southern Africa, MDE: the Middle East, SEA: Southeast Asia, CAS: Central Asia, SAS: South Asia, EAS: East Asia, RBU: Russia/Belarus/Ukraine, PAN: Pacific/Australia/New Zealand and ROW: rest of the world) and (b) the respective percentage contributions to global annual mean combustion $SO_2$ emissions (anthropogenic + biomass burning) from the individual source regions.




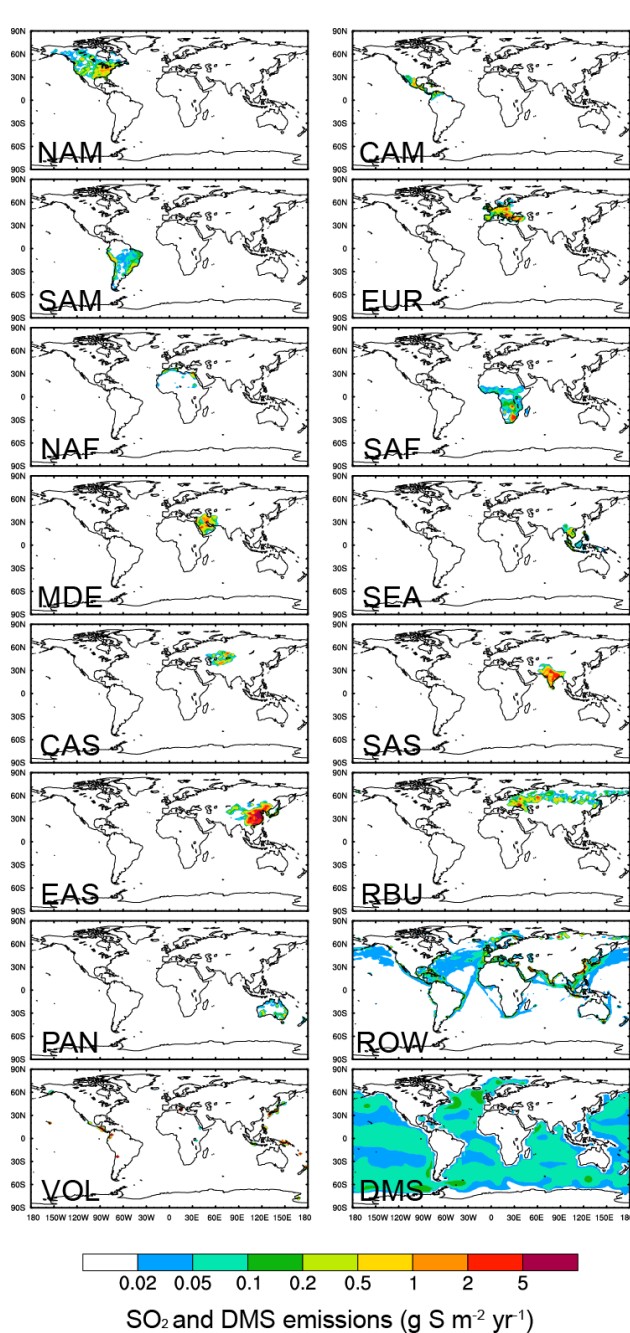



**Figure 2.** Spatial distribution of annual mean emissions (g S m$^{-2}$ yr$^{-1}$) of
anthropogenic SO$_2$, volcanic SO$_2$, and DMS from the sixteen tagged source
regions/sectors averaged over 2010–2014.





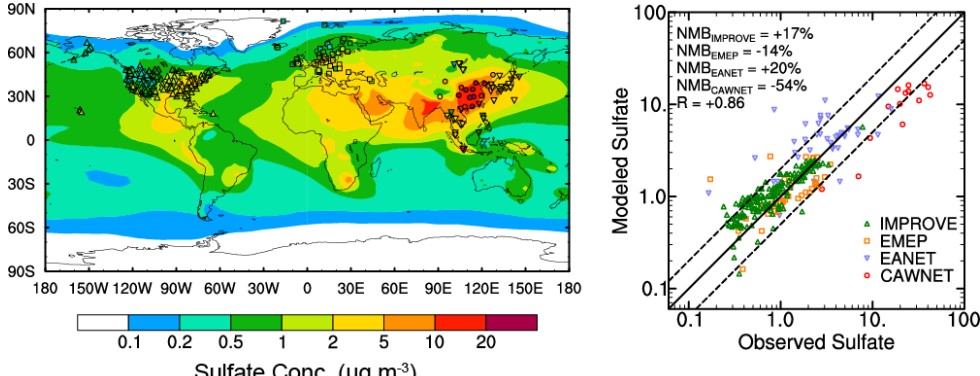

**Figure 3.** Spatial distribution (left panel) and scatter plot (right) between the simulated and observed annual mean near-surface sulfate concentrations (μg m$^{-3}$) over years 2010–2014. Observations are from IMPROVE (up pointing triangle), EMEP (square), EANET (down pointing triangle) for years 2010–2014, and CAWNET (circle) for years 2006–2007, which are scaled to 2010–2014 based on the ratio of CEDS 2010-2014 SO$_2$ emissions to 2006-2007 emissions over China (which is 0.92). Solid lines mark the 1:1 ratio and dashed lines mark the 1:2 and 2:1 ratio. Normalized mean bias (NMB) and correlation coefficient (R) between observation and simulation are shown on the right panel. NMB = 100%$\times\sum(M_i - O_i)/\sum O_i$, where $M_i$ and $O_i$ are the modeled and observed values at site $i$, respectively.





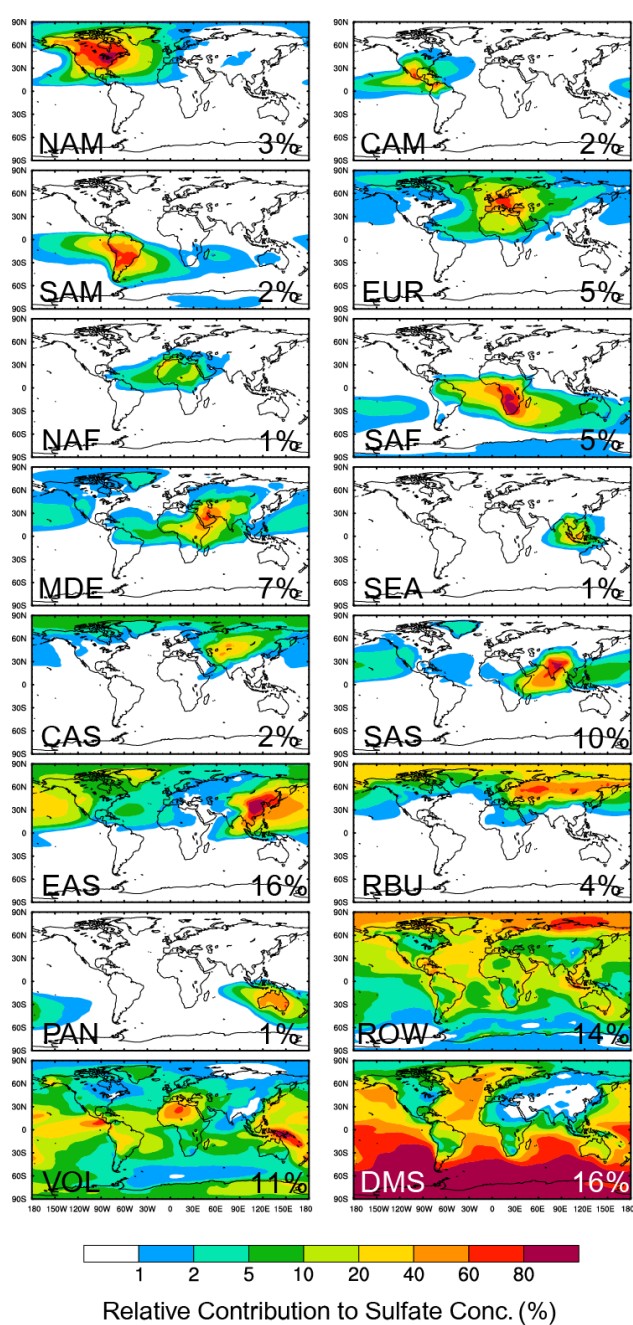

**Figure 4.** Spatial distribution of relative contributions (%) to annual mean
near-surface sulfate concentrations from each of the tagged source regions/sectors.
Relative contributions to global averaged sulfate from individual source
regions/sectors is shown at the bottom right of each panel.





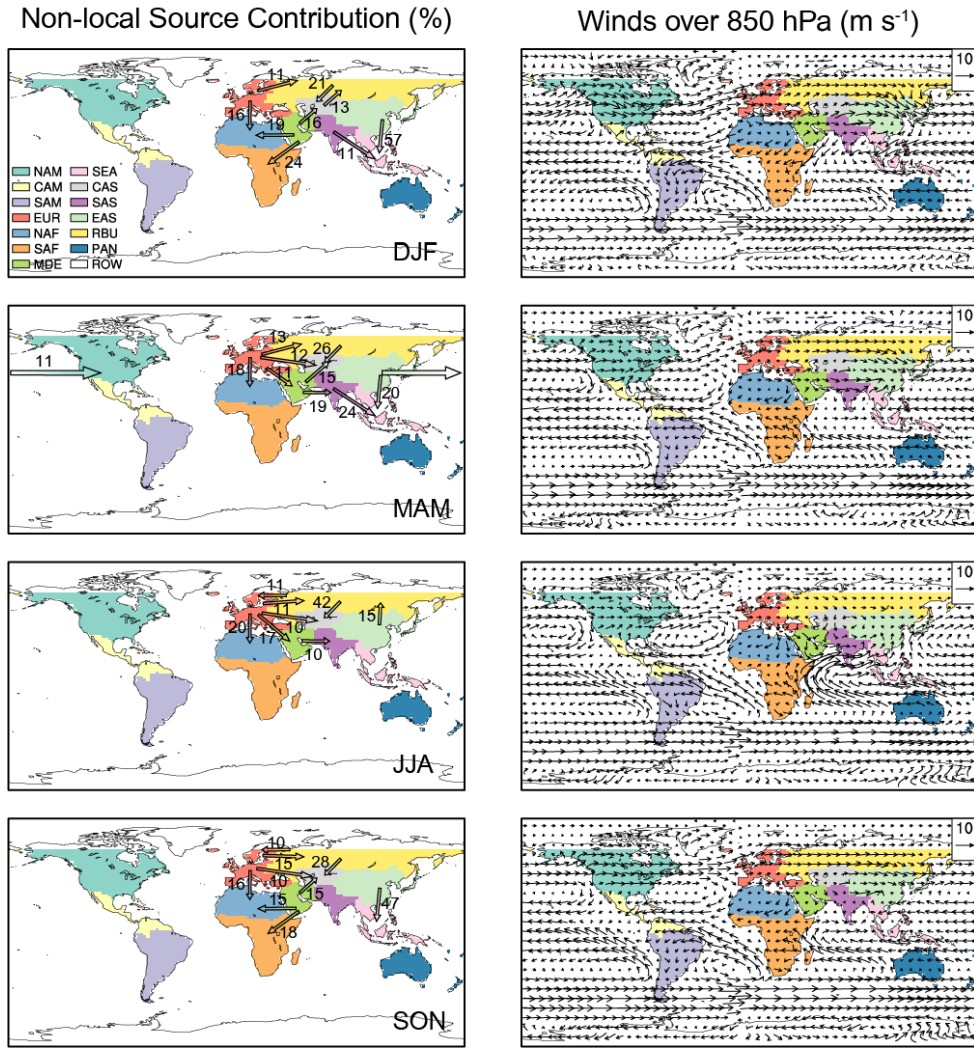

**Figure 5.** Relative contributions of non-local sources to seasonal near-surface sulfate
concentrations (left panels) and wind fields over 850 hPa (right panels). Arrows with
numbers show contributions (%) of a source region to sulfate over a receptor region.
Only relative concentrations larger than 10% are shown.





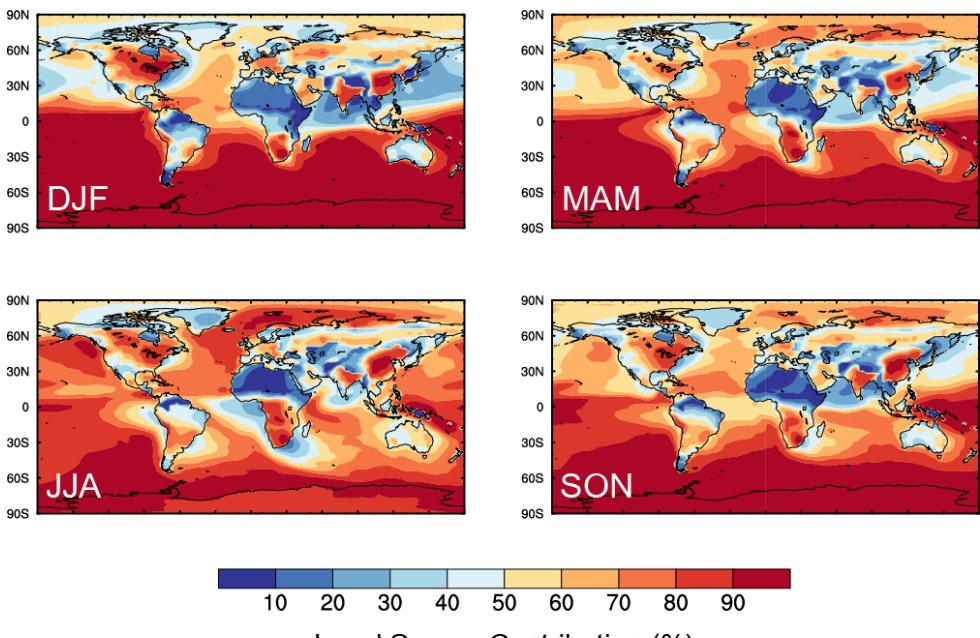



**Figure 6.** Relative contributions (%) of local emissions (inside the tagged regions) to
near-surface sulfate concentrations. Contributions from natural source sectors are
added to ROW here. Contributions less than 50% are shown in cold colors and those
larger than 50% are shown in warm colors.





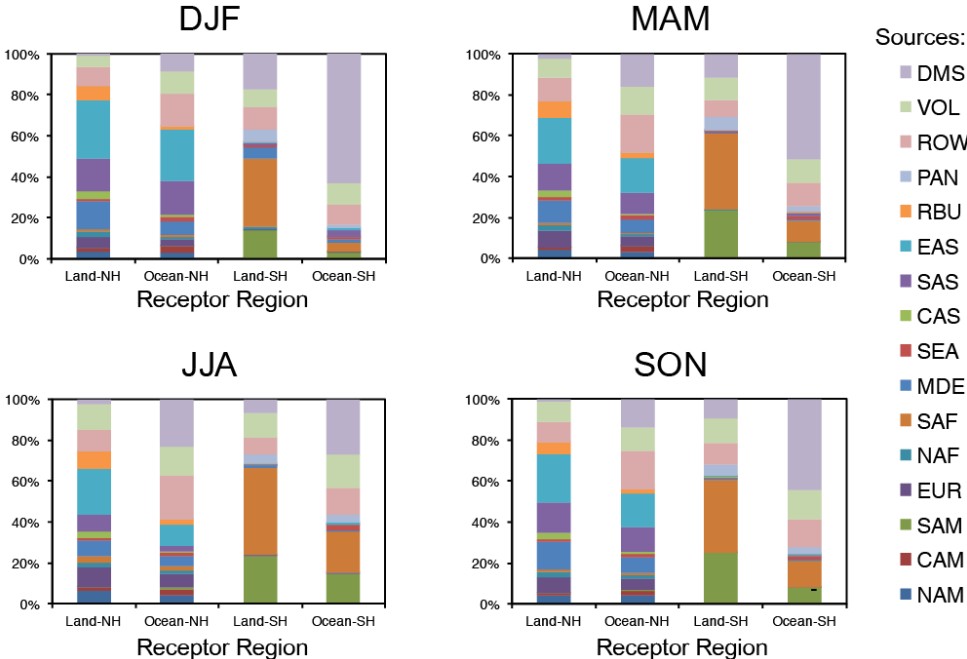

**Figure 7.** Relative contributions (%) to near-surface sulfate concentrations averaged
over land and ocean of the Northern and Southern Hemisphere from emissions in the
sixteen tagged source regions/sectors.





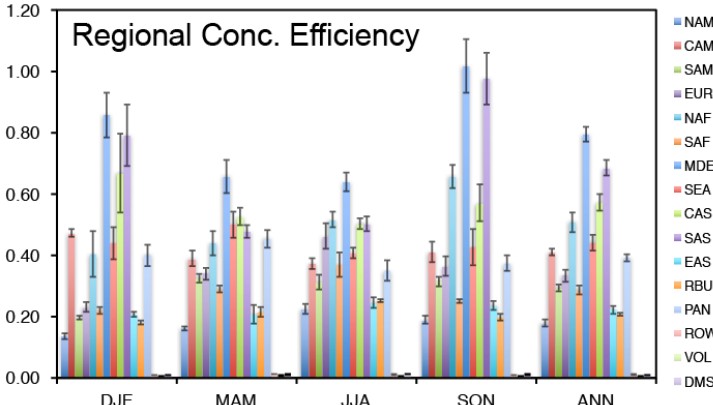

**Figure 8.** Seasonal and annual mean regional concentration efficiency of sulfate (µg m$^{-3}$ (Tg S yr$^{-1}$)$^{-1}$) of the sixteen tagged source regions/sectors. The efficiency is defined as the local contribution to near-surface sulfate concentration divided by the corresponding sulfur emissions from that region (seasonal emissions multiplied by 4). Error bars indicate 1-σ of mean values during years 2010–2014. The receptor region of ROW is used to calculate efficiency of VOL and DMS.



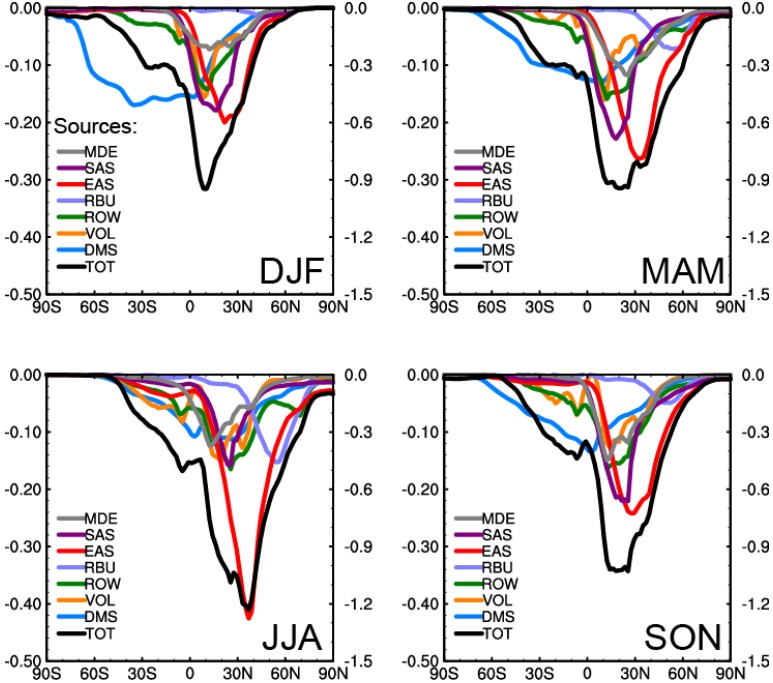

**Figure 9.** Contributions to zonal mean sulfate direct radiative forcing (W m$^{-2}$) from emissions of the tagged regions/sectors shown in colors (left Y axis) and from global total emissions shown in black (right Y axis). Only regions with maximum of zonal mean sulfate direct radiative forcing stronger than –0.1 W m$^{-2}$ are shown here.





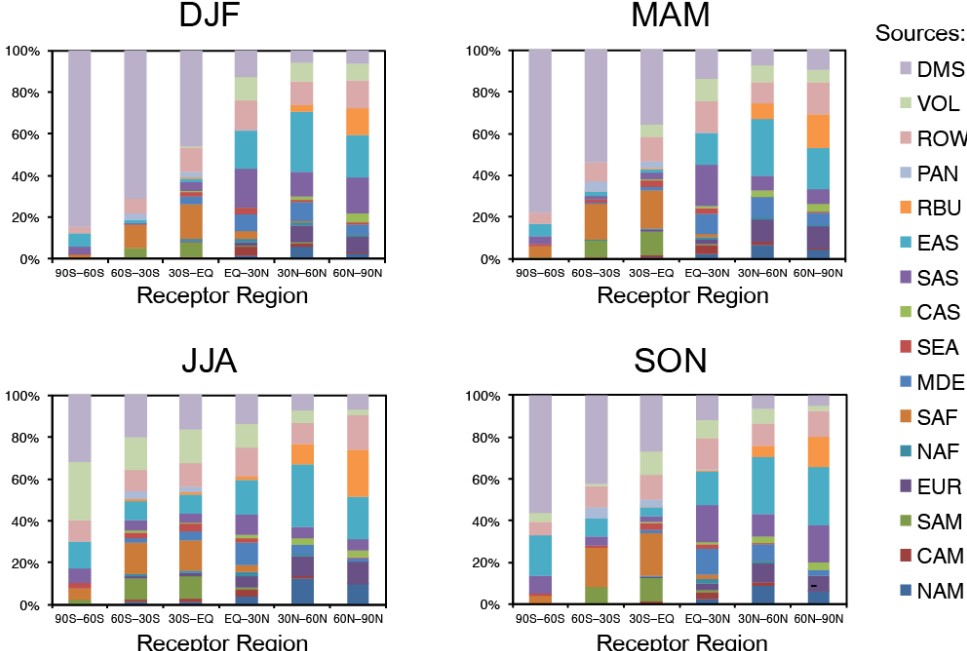

**Figure 10.** Relative contributions (%) from emissions in the sixteen tagged regions/sectors to sulfate direct radiative forcing over the Southern Hemisphere high-latitudes (90°S–60°S), Southern Hemisphere mid-latitudes (60°S–30°S), Southern Hemisphere tropics (30°S–Equator), Northern Hemisphere tropics (Equator–30°N), Northern Hemisphere mid-latitudes (30°N –60°N), and Northern Hemisphere high-latitudes (60°N –90°N).



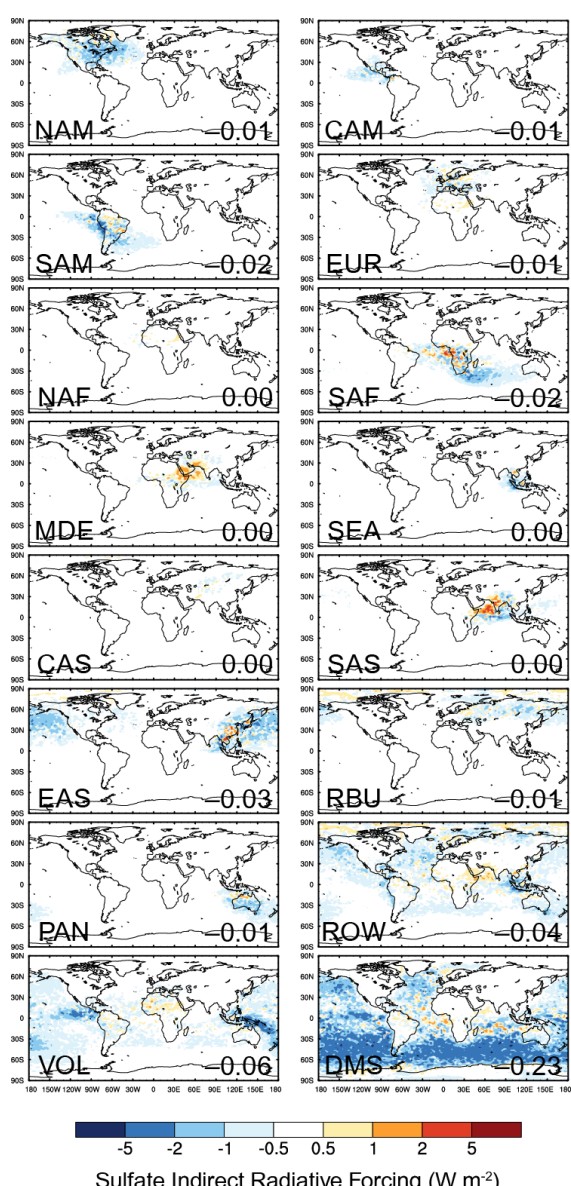

**Figure 11.** Spatial distribution of responses of annual mean indirect radiative forcing of sulfate (IRF, W m$^{-2}$) to a 20% reduction in sulfur emissions (standard simulation – simulation with 20% emission reduction). Regional contributions are calculated as a scaled total incremental IRF in each grid cell by the ratio of source contribution to total sulfate mass concentration reduction averaged from the surface layer to 850 hPa. Regional mean contributions to global incremental IRF of sulfate are shown at the bottom right of each panel.





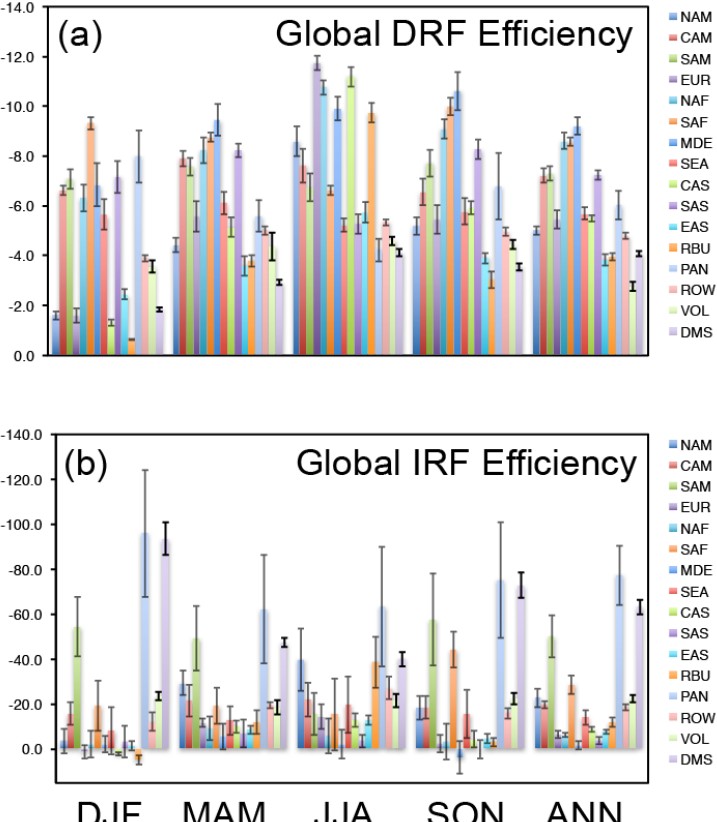



**Figure 12.** Seasonal and annual mean global sulfate (a) direct and (b) indirect
radiative forcing efficiency (mW m$^{-2}$ (Tg S yr$^{-1}$)$^{-1}$) of the sixteen tagged source
regions/sectors. The sulfate radiative efficiency is defined as the global sulfate
radiative forcing divided by the corresponding scaled annual sulfur emission
(seasonal emission multiplied by 4). Error bars indicate 1-σ of mean values during
years 2010–2014.