# Peer review of "Global source attribution of sulfate concentration, direct and # indirect radiative forcing"

_Atmospheric Chemistry and Physics, 2017_

## Referee Comment (RC1) · Anonymous Referee #1 · 3 May 2017

This study quantifies global source-receptor relationships of concentration, direct and indirect radiative forcing of sulfate aerosols utilizing an online chemistry-climate model but nudging it with reanalysis winds. They found that sulfate concentrations are mainly local origin in polluted regions, and their concentration efficiencies in terms of unit precursor emissions are high over arid regions with weak export. In addition, they found the indirect radiative forcing of sulfate aerosols is much larger than the direct radiative forcing. I found topic of this paper is interesting and is suitable for publication in this journal. However, substantial improvements are needed before publication. Following are the major and specific issues:

Major issues:

1.The authors should articulate the novelty or advance in science or methodology of

this study when comparing to previous works. In the introduction, the authors listed a number of similar studies. However, the authors did not describe clearly their motivations to repeat this kind of work as well as the uniqueness of their findings.

2.It is unnecessary to discuss the source-receptor relationships in detail since previous works have already reported similar results. These discussions are lengthy and should be shortened substantially (i.e., abstract, sections 4 and 5). Some figures and discussions could be put into the supporting information.

3.For the method section, the authors may divide it into several subsections (e.g., model description, tracer tagging, model configurations, . . .). In addition, the parameterizations of calculating the DRF and IRF of sulfate need to be described in detail. The method used to calculate the DRF of sulfate from the tagged regions/sectors is also unclear.

4.An incremental IRF is defined in this study to quantify the indirect radiative forcing of sulfate. However, there is no validation about this calculation. As the authors mentioned, anthropogenic sources contributed substantially to the incremental IRF over oceans, but few measurements over remote oceans were used to validate their sulfate calculation. The authors may use some aircraft measurements to verify their results over those remote regions.

5.In the introduction, the authors have mentioned that numerous previous studies have examined the sulfate radiative forcing from different sources and regions. However, in the discussion section, they did not carefully compare their results to previous works. I would suggest the authors pay more attention to the difference between this study and previous works.

Specific commentsïijŽ

1.L139-145: the description about parameterizations and approach that were used to calculate the DRF and IRF is not very clear. Please provide more details.

2.L157: black carbon only occurs in the accumulation mode in MAM3, so the comparison is meaningless.

3.L162: Please show some details about this validation.

4.L198: It is not necessary to show the spatial distributions of SO2 emissions from each tagged region individually. May put Figure 2 into supporting information.

5.L203-210: Need some explanations about these seasonal variations.

6.L219: Only North America is used to validate the decomposition of global incremental IRF. Since different regions may have distinct chemical composition and meteorology, and the sensitivity to regional sulfur emissions could vary significantly by region. I think the authors should validate more regions, especially those with large SO2 emissions, e.g., East Asia, Europe and South Asia.

7.L267: Sulfate has a longer lifetime than black carbon? Need a reference.

8.L282-288: If this bias came from the retrieval algorithm, why this overestimation happened more significantly in China than other regions?

9.L294: Here the model results indicated that the export of SO2 from China is underestimated. However, on Line 291, the authors stated that the inconsistency between simulated results and satellite observations may suggest an overestimation of SO2 at higher altitude. In general, the transport is more efficient in the free troposphere, therefore this indicates a potential overestimation of exporting SO2 from China. Moreover, I would suggest the authors validate total sulfur (SO2+SO4) concentrations and total precipitations over China and downwind region.

10.Sections 4 and 5 are too long and need to be shortened. The authors should pay more attention to the major advance (or unique findings) of this study and explain the difference between their results and previous works.

11.L423: In Table S3, why is the concentration efficiency of sulfate over MDE in SON

greater than 1?

12.L522-525: The sensitivity test with a 20% reduction in regional sulfur emissions over North America indicated a large uncertainty associated with this method. Therefore, I would suggest the authors to discuss more on the uncertainties of this calculation.
* * *

---

## Referee Comment (RC2) · Anonymous Referee #2 · 30 May 2017

The manuscript by Y. Yang et al. examines the source of sulfate concentration and its direct and indirect radiative effect based on a novel source-receptor analysis technology embedded in CESM. Sources from both anthropogenic and natural emissions are identified for different regions over the globe. The model results comply with the expectation from common knowledge and provide qualitative and comprehensive understanding. This research addresses an important and interesting question of where the sulfate aerosol comes from globally and provides some implication for pollution alleviation. But in terms of scientific significance I would not rank this research in the highest catalog because this method has been used in previous studies (Wang et al., 2013; Yang et al., 2017) with different chemical species and regions. Considering that this research provide large amount of detailed and solid analysis that improves our knowledge on the question, I would like to recommend the publication of this manuscript.

[Figure]

Some comments are given below Line 151: Tagging sulfate is "for the first time", however, not the first time used in CESM. Suggest not emphasizing the novelty. Line 156 to 159: The same reason as above. Suggest just elaborate the method and avoid using phrases such as "In contrast". Line 215: why using 20% reduction to evaluate the indirect effect? Although 20% reduction was used in a previous study (Stjern et al., 2016), this increment of emission is arbitrary to me. Moreover, it hinders comparison with the magnitude of DRF, which compares the forcing with and without 100% aerosols. Line 449: is this 1% difference a coincidence?
* * *

---

## Author Comment (AC1) · 19 Jun 2017

**Manuscript # acp-2017-303**

**Responses to Reviewer #1**

This study quantifies global source-receptor relationships of concentration, direct and indirect radiative forcing of sulfate aerosols utilizing an online chemistry-climate model but nudging it with reanalysis winds. They found that sulfate concentrations are mainly local origin in polluted regions, and their concentration efficiencies in terms of unit precursor emissions are high over arid regions with weak export. In addition, they found the indirect radiative forcing of sulfate aerosols is much larger than the direct radiative forcing. I found topic of this paper is interesting and is suitable for publication in this journal. However, substantial improvements are needed before publication. Following are the major and specific issues:

We thank the referee for all the insightful comments to the manuscript and helpful suggestions for improving the presentation quality. Below, we explain how the comments and suggestions are addressed (our point-by-point responses in blue) and make note of the changes that have been made to the manuscript, attempting to take into account all the comments raised here.

Major issues:

1.The authors should articulate the novelty or advance in science or methodology of this study when comparing to previous works. In the introduction, the authors listed a number of similar studies. However, the authors did not describe clearly their motivations to repeat this kind of work as well as the uniqueness of their findings.
Response:
In previous studies about source attribution of sulfate, only a limited number of anthropogenic source regions over the Northern Hemisphere were considered and examined (Park et al., 2004; Heald et al., 2006; Chin et al., 2007; Hadley et al., 2007; Yu et al., 2013; Bellouin et al., 2016; Stjern et al., 2016). Continents and subcontinents over the tropics and Southern Hemisphere are also important source and receptor regions for the sulfate radiative forcing, especially indirect forcing due to stronger aerosol-cloud interactions in clean environments (Koren et al., 2014). Although Liu and Mauzerall (2007) and Liu et al. (2008, 2009) included ten anthropogenic source regions, they only focused on source attribution of sulfate mass concentration without examining sulfate radiative forcing. In addition, few studies have quantified the global source-receptor relationships of sulfate indirect radiative forcing that can be attributed to local/non-local source regions and anthropogenic/natural source sectors.

Certainly, this study is not a repeat of previous work on source-receptor relationships. This is the first study that examines source attribution of sulfate radiative forcing with tagged anthropogenic and natural sources covering the whole globe. In this study, we quantify sixteen source region/sector contributions (fourteen major source regions and two natural source sectors) to regional and global sulfate mass concentrations, and direct and indirect radiative forcing of sulfate. Another novel aspect of this study is that we are using the new emissions datasets generated for the CMIP6 activities. Thus our model configuration and results could potentially be more comparable to future modeling results coming out of the CMIP6 activities than most of the previous studies.

We have revised the introduction section to show these novelties and discussed the differences between our study and previous studies. Please see responses to the more specific comments below.

2.It is unnecessary to discuss the source-receptor relationships in detail since previous works have already reported similar results. These discussions are lengthy and should be shortened substantially (i.e., abstract, sections 4 and 5). Some figures and discussions could be put into the supporting information.
Response:

We have significantly shortened the details of source-receptor relationships in the abstract as suggested. We would like to stress that our systematical analysis is from a variety of angles to describe source-receptor relationships of sulfate concentrations and radiative forcing, including both the near surface concentration and column burden, both direct and indirect radiative forcing, both oceanic and continental regions, both Northern and Southern Hemisphere, both anthropogenic and natural sources, and both absolute and relative contributions. For all these aspects, we don't see overall similar results to previous studies, especially, in a quantitative way. However, we do value the comment on our lengthy discussions, and have tried our best to shorten the source-receptor descriptions in sections 4 and 5.

3.For the method section, the authors may divide it into several subsections (e.g., model description, tracer tagging, model configurations, . . .). In addition, the parameterizations of calculating the DRF and IRF of sulfate need to be described in detail. The method used to calculate the DRF of sulfate from the tagged regions/sectors is also unclear.
Response:

We have now divided it into the suggested subsections: model description, sulfur source-tagging, emissions, and model configurations. We also clarified on the configurations of the list of simulations performed in this study.

We have also revised the description of parameterizations and the approach of calculating DRF and IRF to include more details, as the following:
"Sulfate is internally mixed with other species in the same aerosol mode and

then externally mixed between modes. Sulfate refractive indices at visible wavelengths is 1.43+0.00i. Activation of cloud droplets uses the scheme from Abdul-Razzak and Ghan (2000). The model simulates aerosol-cloud interactions in stratiform clouds using a physically based two-moment parameterization (Morrison and Gettelman, 2008). In addition to the standard radiative fluxes calculated in the model by taking into account all aerosols, the CESM has the capability of diagnosing radiative fluxes in parallel for a subset of aerosol species. The difference between the standard and the diagnosed radiative fluxes can then be attributed to the difference in aerosols considered in the radiation calculations. For example, the difference in shortwave radiation fluxes at the top of the atmosphere (TOA) represents the DRF of the excluded aerosol components in the diagnostic calculation (Ghan, 2013). Using this same method, the DRF of sulfate from any of the sixteen individual tagged regions/sectors can be derived from a pair of diagnostic radiation calculations with and without the particular tagged sulfate considered. To estimate IRF of sulfate from different sources, we define in this study an incremental IRF, calculated as $\Delta(F_{clean} - F_{clear,clean})$, where F is the radiative flux at TOA, $F_{clean}$ is the flux calculated neglecting scattering and absorption by aerosols, $F_{clear,clean}$ is the flux calculated neglecting scattering and absorption by both clouds and aerosols, and $\Delta$ refers to the differences between the base and emission perturbed simulations."

4. An incremental IRF is defined in this study to quantify the indirect radiative forcing of sulfate. However, there is no validation about this calculation. As the authors mentioned, anthropogenic sources contributed substantially to the incremental IRF over oceans, but few measurements over remote oceans were used to validate their sulfate calculation. The authors may use some aircraft measurements to verify their results over those remote regions.
Response:
    The sulfate indirect effect has been fully validated in McCoy et al. (2017) with the same model (CAM5.1-MAM3-PNNL in their study). McCoy et al. (2017) reported that the CAM5.1-MAM3-PNNL model did quite well at producing a reasonable sensitivity of cloud to sulfate mass concentration compared to MODIS satellite data. In addition, in another multi-model intercomparison study including the base simulation results from this work, Fanourgakis et al. (in preparation) evaluates aerosol, CCN and cloud sensitivity in global models against several observational datasets. The incremental IRF in this study is also derived based on the sensitivity of cloud forcing to sulfate (20% of sulfate precursor emission). We have now cited these studies in the revised manuscript instead of duplicating the work.

5. In the introduction, the authors have mentioned that numerous previous studies have examined the sulfate radiative forcing from different sources and regions. However, in the discussion section, they did not carefully compare

their results to previous works. I would suggest the authors pay more attention to the difference between this study and previous works.

Response:

Because we are using different emission datasets and source regions from those in previous studies, a quantitative comparison of source attributions is not so meaningful. However, it is more interesting to compare the radiative forcing efficiency with previous studies. We have now added the Table S9 (see below) to show the comparisons and discussed it at the last section of the manuscript, as the following:

"Table S9 compares the annual sulfate radiative forcing efficiencies simulated in this study to those in previous multi-model studies (Yu et al., 2013; Bellouin et al., 2016; Stjern et al., 2016). As in the previous studies, the DRF efficiency is calculated as the response of global DRF to a 20% reduction in local emissions divided by the 20% of sulfur emissions based on two separate simulations rather than 100% of local emissions in a single simulation (Table S6). The efficiencies based on the 20% emission reduction are very similar to those of the 100% emission reduction, indicating a nearly linear relationship between sulfate DRF and emissions. Compared to Yu et al. (2013) and Stjern et al. (2016), the DRF efficiencies in this study are around the lower bound for all source regions. Another multi-model intercomparison study also reported a lower sulfate DRF simulated in CAM5 compared to other models (Myhre et al., 2013). The difference in DRF efficiencies likely arises from differences in the estimates of aerosol optical properties. With aerosol-cloud interactions included, the total radiative forcing efficiencies in this study are similar to the best estimates provided by Bellouin et al. (2016). The global IRF in CAM5 was also found to be larger than other models in a nine-model intercomparison study, which was attributed to an strong aerosol induced cloud scattering (Zelinka et al., 2014)."

**Table S9.** Comparison of annual sulfate radiative forcing efficiency (mW m$^{-2}$ (Tg S yr$^{-1}$)$^{-1}$) in this study and previous studies. The sulfate DRF efficiencies are calculated as the response of global DRF to a 20% reduction in local emissions divided by the 20% of sulfur emissions.

| | EUR | EAS | NAM | SAS | RBU | MDE |
|---|---|---|---|---|---|---|
| Direct radiative forcing (DRF) efficiency | | | | | | |
| Yu et al. (2013) | −9.8~−5.0 | −7.6~−3.2 | −10.0~−5.0 | −10.8~−5.0 | | |
| Stjern et al. (2016) | −15.7~−5.6 | −12.1~−4.6 | −15.5~−4.1 | −28.0~−6.3 | −8.9~−4.3 | −32.4~−10.9 |
| This study | −5.4 | −3.8 | −4.8 | −7.2 | −3.9 | −9.4 |
| Total (direct + indirect) radiative forcing efficiency | | | | | | |
| | EUR | EAS | | | | |
| Bellouin et al. (2016) | −13.0 (−22.7~−4.4) | −9.5(−13.6~−2.6) | | | | |
| This study | −12.0 | −11.6 | | | | |

Specific comments:

1.L139-145: the description about parameterizations and approach that were used to calculate the DRF and IRF is not very clear. Please provide more details.
Response:
    We have revised the description. Please see the response to comment #3 of major issues.

2.L157: black carbon only occurs in the accumulation mode in MAM3, so the comparison is meaningless.
Response:
    We have deleted this sentence.

3.L162: Please show some details about this validation.
Response:
    We have added Fig. S1 to compare the sulfate concentration and surface air temperature between the no-tagging and tagging simulations to validate the sulfur tagging technique used in this study.

[Figure]

**Figure S1.** Spatial distribution of annual mean near-surface sulfate concentrations (left, μg m$^{-3}$) and surface air temperature (right, K) from no-tagging (top), tagging (middle) simulations and their differences (bottom).

4.L198: It is not necessary to show the spatial distributions of SO2 emissions from each tagged region individually. May put Figure 2 into supporting information.
Response:
    We have moved this figure to the supporting information as suggested.

5.L203-210: Need some explanations about these seasonal variations.
Response:
    We have added some explanations, as "East Asia, RBU and Europe have seasonal peak emissions in boreal winter due to high residential emissions from heating in this season together with higher SO$_2$ emission from the energy sector. Southern Africa shows larger emission in boreal summer from biomass burning in this season, while emissions from North America are comparable in winter and summer. DMS is emitted over oceans with a boreal winter peak due to phytoplankton blooms over the Southern Ocean."

6.L219: Only North America is used to validate the decomposition of global incremental IRF. Since different regions may have distinct chemical composition and meteorology, and the sensitivity to regional sulfur emissions could vary significantly by region. I think the authors should validate more regions, especially those with large SO2 emissions, e.g., East Asia, Europe and South Asia.
Response:
    We agree with the reviewer that using only North America to validate the decomposition of global incremental IRF may not be sufficient. However, it is computationally infeasible to test many of the source regions. Given the large emissions from East Asia, we also performed an additional sensitivity simulation with a 20% reduction in regional sulfur emissions over East Asia and have added the IRF comparison in Fig. S10, with results also now included in the text.
    We have also revised the description of the comparison as "The 20% emission from North America results in negative IRF over Eastern U.S. and downwind ocean regions. The 20% emission in East Asia emissions produces negative IRF over the northwestern Pacific. Globally, DMS, North America and East Asia contribute to –0.230 (±0.012), –0.014 (±0.002), and –0.028 (±0.003) W m$^{-2}$, respectively, of sulfate incremental IRF from the method with sulfur tagging technique, similar to –0.248 (±0.020), –0.018 (±0.019), and –0.028 (±0.018) W m$^{-2}$, from the individual emission-perturbation simulations."
    We have also added a discussion of the noisy spatial distribution of IRF in the emission perturbation method shown in comment #12.

[Figure]

**Figure S10.** Spatial distribution of annual mean IRF of sulfate (W m$^{-2}$) induced by a 20% reduction in sulfur emissions from the decomposition using the sulfur tagging method (left panels) and a simple 20% regional/sector emission perturbation (right) for source from DMS (top panels), North America (middle panels), and East Asia (bottom panels).

7.L267: Sulfate has a longer lifetime than black carbon? Need a reference.
Response:

We were thinking about the additional time for the gas-to-particle conversion. It seems to cause some confusion, so we have deleted this sentence.

8.L282-288: If this bias came from the retrieval algorithm, why this overestimation happened more significantly in China than other regions?
Response:

Not only in China, the simulated SO$_2$ burden is 3 times larger than OMI data over North America, 7 times over Europe, and 5 times over Southeast Asia. We have added these in the manuscript.

9.L294: Here the model results indicated that the export of SO2 from China is under-estimated. However, on Line 291, the authors stated that the inconsistency between simulated results and satellite observations may suggest an overestimation of SO2 at higher altitude. In general, the transport is more efficient in the free troposphere, therefore this indicates a potential overestimation of exporting SO2 from China. Moreover, I would suggest the authors validate total sulfur (SO2+SO4) concentrations and total precipitations over China and downwind region.

Response:

Observational data of $SO_2$ and sulfate are from different sites and have different time coverage. It is difficult to validate total sulfur over China. Nonetheless, considering that both $SO_2$ and sulfate are underestimated in the model compared to site observations, the total sulfur is likely to be underestimated. We have added Fig. S5 to validate total precipitation. Over China, CAM5 overestimates precipitation over northern China, which leads to a strong aerosol scavenging and low sulfate concentration over this region. We have added these in the manuscript.

We have also revised the discussion of model-observation comparison for clarification, as "The simulated near-surface $SO_2$ concentrations, however, are also underestimated by 25% compared to observations over thirteen sites in China (Gong et al., 2014) shown in Fig. S4a, also suggesting a large bias in satellite retrievals or too much $SO_2$ simulated in higher altitude. In general, the transport is more efficient in the free troposphere. If too much $SO_2$ is simulated in higher altitude, the near-surface $SO_2$ concentration is likely to be overestimated over downwind regions. However, the modeled $SO_2$ concentrations over downwind regions of China are underestimated by 45% compared to observations from EANET sites (Fig. S4b). This indicates that bias in the satellite retrievals may be a significant cause of the inconsistency between modeled and satellite-estimated SO2 burden."

[Figure]

**Figure S5.** Spatial distribution of annual mean precipitation (mm day$^{-1}$) from CMAP (Climate Prediction Center's Merged Analysis of Precipitation, top) and simulated in this study (bottom) averaged over 2010–2014.

10.Sections 4 and 5 are too long and need to be shortened. The authors should pay more attention to the major advance (or unique findings) of this study and explain the difference between their results and previous works.
Response:

We have tried our best to shorten the source-receptor description in sections 4 and 5. As we explained in the response to the major comment above, we don't necessarily expect similar results to previous studies. As shown in the newly added Table S9, previous studies only examined influence from limited source regions (2–6). In this study, we have 16 tagged source regions partly owing to the computationally efficient sulfur tagging technique, which extends the source-receptor relationship to the whole globe. For the comparison to limited source regions examined in previous studies, we have discussed the differences and possible biases of the model in response to comment #5 of major issues.

11.L423: In Table S3, why is the concentration efficiency of sulfate over MDE in SON greater than 1?
Response:

The efficiencies over the Middle East show high values in almost all seasons due to dry atmospheric conditions favoring long aerosol lifetime,

especially in DJF and SON. We have emphasized it in the manuscript. The concentration efficiency is calculated as local contribution to the near-surface sulfate concentration divided by local sulfur emission (seasonal emissions multiplied by 4). In SON, The MDE local contribution to concentration is 3.40 $\mu g$ $m^{-3}$ and its local $SO_2$ emission is 0.835 Tg S $yr^{-1}$. Efficiency=3.40/(0.835*4)=1.02 $\mu g$ $m^{-3}$ (Tg S $yr^{-1}$)$^{-1}$). Since the efficiency is not normalized, it does not have to be less than 1.

12.L522-525: The sensitivity test with a 20% reduction in regional sulfur emissions over North America indicated a large uncertainty associated with this method. Therefore, I would suggest the authors to discuss more on the uncertainties of this calculation.
Response:

Thanks for the suggestion. We have added a discussion of this large uncertainty, as "The latter method has larger noise, seen in both the spatial distributions and large uncertainties (standard deviation) of the incremental IRF. The three emission-perturbed simulations produced similar system noise, with a magnitude of ~0.02 W $m^{-2}$. The incremental IRF signal is larger than the noise around the source regions whereas noise masks the signal in other regions, leading to large uncertainties. However, in the simulation with all source emissions reduced by 20%, the IRF signal overwhelms noise almost everywhere. With the sulfur tagging technique and decomposition method, the noise is also decomposed into smaller pieces which are, in turn, much smaller than the decomposed incremental IRF signal."

References:

[revised manuscript text omitted]

---

## Author Comment (AC2) · 19 Jun 2017

**Manuscript # acp-2017-303**

**Responses to Reviewer #2**

The manuscript by Y. Yang et al. examines the source of sulfate concentration and its direct and indirect radiative effect based on a novel source-receptor analysis technology embedded in CESM. Sources from both anthropogenic and natural emissions are identified for different regions over the globe. The model results comply with the expectation from common knowledge and provide qualitative and comprehensive understanding. This research addresses an important and interesting question of where the sulfate aerosol comes from globally and provides some implication for pollution alleviation. But in terms of scientific significance I would not rank this research in the highest catalog because this method has been used in previous studies (Wang et al., 2013; Yang et al., 2017) with different chemical species and regions. Considering that this research provide large amount of detailed and solid analysis that improves our knowledge on the question, I would like to recommend the publication of this manuscript. Some comments are given below

We thank the referee for all the comments to the manuscript for improving the presentation quality and the recommendation for publication. Regarding the major comment on the relatively low scientific significance, we would argue that the sulfur tagging technique was implemented in the Community Earth System Model (CESM), for the first time, in this study. Our previous studies used a black carbon (BC) tagging method to study the source attributions and impact of BC emitted from different source regions/sectors. The sulfur-tagging and BC-tagging share the same idea, but compared to BC, sulfur has additional gas-phase and aqueous-phase chemical reactions and there are more size-modes to treat sulfate particles. Thus the sulfur-tagging code implementation, testing and validation did take a large amount of additional efforts, and this tool is indeed novel and unique to the present study.

Below, we explain how the comments and suggestions are addressed (our point-by-point responses in blue) and make note of the changes that have been made to the manuscript, attempting to take into account all the comments raised here.

Line 151: Tagging sulfate is "for the first time", however, not the first time used in CESM. Suggest not emphasizing the novelty.
Response:
    Please see our response above. Sulfate tagging is indeed for the first time implemented and used in CESM.

Line 156 to 159: The same reason as above. Suggest just elaborate the method and avoid using phrases such as "In contrast".
Response:
    Thanks for the suggestion. We don't have to contrast it to BC-tagging, so we changed the description and deleted this phrase.

Line 215: why using 20% reduction to evaluate the indirect effect? Although 20% reduction was used in a previous study (Stjern et al., 2016), this increment of emission is arbitrary to me. Moreover, it hinders comparison with the magnitude of DRF, which compares the forcing with and without 100% aerosols.
Response:
    We agree that the 20% reduction of emissions is somewhat arbitrary. However, it follows the AeroCom multi-model experiments design in the framework HTAP (Hemispheric Transport of Air Pollution) to examine the significance of emission reduction. There are numerous studies examined air quality and climate responses to a 20% emission reductions (e.g. Fiore et al., 2009; Fry et al., 2012; Yu et al., 2013; Stjern et al., 2016; Bellouin et al., 2016). We use the same amount of reduction for the purpose of having a fair comparison to these studies. We have added an explanation in the model configuration part, as "The 20% is chosen to follow the experiment design in the framework HTAP2."
    We also agree with the reviewer that the DRF calculated with and without 100% sulfate cannot be directly compared with the incremental IRF induced by the 20% change in emissions. Therefore, we have added in Table S8 the incremental DRF/IRF (calculated from the base and 20% emission reduction simulations) and the standard DRF/IRF (based on the present-day and preindustrial emission simulations), as well as their radiative forcing efficiencies. We have also added a discussion of the comparisons, as "For comparison, Table S8 also includes the incremental DRF calculated with the same simulations for the incremental IRF and the standard anthropogenic DRF between present-day and preindustrial conditions, as well as their efficiencies. The forcing efficiencies are also similar between the incremental and the standard anthropogenic DRF. The IRF and its efficiencies are much higher than those of DRF for sources over or around clean oceanic regions (e.g., DMS, volcanic $SO_2$, emissions from Australia and South America), but much lower for regions with high emissions (e.g., the Middle East, South Asia)."

**Table S8.** Annual sulfate incremental direct and indirect radiative forcing calculated based on simulations with and without 20% reduction in sulfur emissions globally and standard direct and indirect radiative forcing (W m$^{-2}$)

calculated based on simulations using present-day and preindustrial emissions, as well as the forcing efficiencies (mW m$^{-2}$ (Tg S yr$^{-1}$)$^{-1}$) for all of the sixteen tagged source regions/sectors.

| DRF Forcing | | | | | | | |
|---|---|---|---|---|---|---|---|
| | NAM | CAM | SAM | EUR | NAF | SAF | MDE | SEA |
| Incremental DRF | -0.003 | -0.002 | -0.002 | -0.004 | -0.001 | -0.005 | -0.006 | -0.002 |
| DRF (PD–PI) | -0.015 | -0.010 | -0.011 | -0.018 | -0.005 | -0.023 | -0.031 | -0.008 |
| | CAS | SAS | EAS | RBU | PAN | ROW | VOL | DMS |
| Incremental DRF | -0.001 | -0.009 | -0.014 | -0.002 | -0.001 | -0.010 | -0.007 | -0.014 |
| DRF (PD–PI) | -0.006 | -0.046 | -0.068 | -0.011 | -0.003 | -0.053 | | |
| DRF Efficiency | | | | | | | |
| | NAM | CAM | SAM | EUR | NAF | SAF | MDE | SEA |
| Incremental DRF efficiency | -4.8 | -6.9 | -7.1 | -5.4 | -8.3 | -8.4 | -9.4 | -5.3 |
| DRF efficiency | -4.9 | -7.0 | -7.1 | -5.4 | -8.1 | -8.5 | -9.1 | -5.5 |
| | CAS | SAS | EAS | RBU | PAN | ROW | VOL | DMS |
| Incremental DRF efficiency | -5.4 | -7.2 | -3.8 | -3.9 | -5.5 | -4.6 | -2.7 | -4.0 |
| DRF efficiency | -5.2 | -7.2 | -3.8 | -3.9 | -5.5 | -4.8 | | |
| IRF Forcing | | | | | | | |
| | NAM | CAM | SAM | EUR | NAF | SAF | MDE | SEA |
| Incremental IRF | -0.014 | -0.006 | -0.016 | -0.004 | -0.001 | -0.016 | -0.001 | -0.004 |
| IRF (PD–PI) | -0.082 | -0.036 | -0.072 | -0.032 | -0.005 | -0.061 | 0.012 | -0.017 |
| | CAS | SAS | EAS | RBU | PAN | ROW | VOL | DMS |
| Incremental IRF | -0.002 | -0.005 | -0.028 | -0.007 | -0.009 | -0.042 | -0.057 | -0.230 |
| IRF (PD–PI) | -0.012 | -0.002 | -0.117 | -0.056 | -0.051 | -0.202 | | |
| IRF Efficiency | | | | | | | |
| | NAM | CAM | SAM | EUR | NAF | SAF | MDE | SEA |
| Incremental IRF efficiency | -22.8 | -19.8 | -50.3 | -6.6 | -6.2 | -28.7 | -1.7 | -14.1 |
| IRF efficiency | -26.3 | -25.0 | -44.7 | -9.5 | -7.9 | -22.4 | 3.5 | -11.9 |
| | CAS | SAS | EAS | RBU | PAN | ROW | VOL | DMS |
| Incremental IRF efficiency | -8.7 | -3.7 | -7.8 | -11.8 | -77.3 | -18.6 | -22.5 | -63.2 |
| IRF efficiency | -11.5 | -0.3 | -6.6 | -18.7 | -86.6 | -18.1 | | |

Line 449: is this 1% difference a coincidence?
Response:
It does not mean '1% difference', but '1750 emission is less than 1% of present-day emission'. We tried to illustrate that DRF of anthropogenic sulfate is calculated here based on present-day and no emission condition (DRF$_{PD}$ – 0), while the estimate in IPCC AR5 represents the difference between the present-day and 1750 DRF (DRF$_{PD}$ – DRF$_{1750}$). The global anthropogenic SO$_2$ emission amount (0.5 Tg/yr) in 1750 is very small, about 0.5% (less than "**1%**")

of the 2010–2014 level (109.8 Tg/yr) from the CEDS emission dataset. Therefore, $DRF_{PD} - DRF_{1750} \approx DRF_{PD}$.

References:

Fiore, A. M., et al. (2009), Multimodel estimates of intercontinental source-receptor relationships for ozone pollution, J. Geophys. Res., 114, D04301, doi:10.1029/2008JD010816.

Fry, M. M., et al. (2012), The influence of ozone precursor emissions from four world regions on tropospheric composition and radiative climate forcing, J. Geophys. Res., 117, D07306, doi:10.1029/2011JD017134.

Yu, H., Chin, M., West, J. J., Atherton, C. S., Bellouin, N., Bergmann, D., Bey, I., Bian, H., Diehl, T., Forberth, G., Hess, P., Schulz, M., Shindell, D., Takemura, T., and Tan, Q.: A multimodel assessment of the influence of regional anthropogenic emission reductions on aerosol direct radiative forcing and the role of intercontinental transport, J. Geophys. Res. Atmos., 118, 700-720, doi:10.1029/2012JD018148, 2013.

Stjern, C. W., Samset, B. H., Myhre, G., Bian, H., Chin, M., Davila, Y., Dentener, F., Emmons, L., Flemming, J., Haslerud, A. S., Henze, D., Jonson, J. E., Kucsera, T., Lund, M. T., Schulz, M., Sudo, K., Takemura, T., and Tilmes, S.: Global and regional radiative forcing from 20 % reductions in BC, OC and SO4 – an HTAP2 multi-model study, Atmos. Chem. Phys., 16, 13579-13599, doi:10.5194/acp-16-13579-2016, 2016.

Bellouin, N., Baker, L., Hodnebrog, Ø., Olivié, D., Cherian, R., Macintosh, C., Samset, B., Esteve, A., Aamaas, B., Quaas, J., and Myhre, G.: Regional and seasonal radiative forcing by perturbations to aerosol and ozone precursor emissions, Atmos. Chem. Phys., 16, 13885-13910, doi:10.5194/acp-16-13885-2016, 2016.